# Spatiotemporal manipulation of ciliary glutamylation reveals its roles in intraciliary trafficking and Hedgehog signaling

Shi-Rong Hong[1], Cuei-Ling Wang[1], Yao-Shen Huang[1], Yu-Chen Chang[2], Ya-Chu Chang[1], Ganesh V. Pusapati[3], Chun-Yu Lin[4], Ning Hsu[1], Hsiao-Chi Cheng[4], Yueh-Chen Chiang[5], Wei-En Huang[2], Nathan C. Shaner[6], Rajat Rohatgi [3], Takanari Inoue[7] & Yu-Chun Lin[1,4]

Tubulin post-translational modifications (PTMs) occur spatiotemporally throughout cells and are suggested to be involved in a wide range of cellular activities. However, the complexity and dynamic distribution of tubulin PTMs within cells have hindered the understanding of their physiological roles in specific subcellular compartments. Here, we develop a method to rapidly deplete tubulin glutamylation inside the primary cilia, a microtubule-based sensory organelle protruding on the cell surface, by targeting an engineered deglutamylase to the cilia in minutes. This rapid deglutamylation quickly leads to altered ciliary functions such as kinesin-2-mediated anterograde intraflagellar transport and Hedgehog signaling, along with no apparent crosstalk to other PTMs such as acetylation and detyrosination. Our study offers a feasible approach to spatiotemporally manipulate tubulin PTMs in living cells. Future expansion of the repertoire of actuators that regulate PTMs may facilitate a comprehensive understanding of how diverse tubulin PTMs encode ciliary as well as cellular functions.

[1] Institute of Molecular Medicine, National Tsing Hua University, Hsinchu 30013, Taiwan. [2] Department of Life Science, National Tsing Hua University, Hsinchu 30013, Taiwan. [3] Departments of Medicine and Biochemistry, Stanford University School of Medicine, Stanford 94305 CA, USA. [4] Department of Medical Science, National Tsing Hua University, Hsinchu 30013, Taiwan. [5] Interdisciplinary Program of Science, National Tsing Hua University, Hsinchu 30013, Taiwan. [6] Department of Photobiology and Bioimaging, The Scintillon Institute, San Diego 92121 CA, USA. [7] Department of Cell Biology, Center for Cell Dynamics, School of Medicine, Johns Hopkins University, Baltimore 21205 MD, USA. These authors contributed equally: Shi-Rong Hong and Cuei-Ling Wang. Correspondence and requests for materials should be addressed to T.I. (email: jctinoue@jhmi.edu) or to Y.-C.L. (email: ycl@life.nthu.edu.tw)

The primary cilium is a microtubule-based sensory organelle protruding from the apical surface of resting cells; it is crucial in phototransduction, olfaction, hearing, embryonic development, and several cellular-signaling pathways, such as Hedgehog (Hh) signaling[1, 2]. Defects in primary cilia lead to a number of human diseases[3]. Structurally, the cilium is composed of nine microtubule doublets called the axoneme, which offer mechanical support to the cilium, and also provide tracks for motor protein-dependent trafficking, known as intraflagellar transport (IFT)[4]. Polyglutamylation generates glutamate chains of varying lengths at the C-terminal tails of axonemal tubulin[5, 6]. This post-translational modification (PTM) occurs on the surface of microtubules and provides interacting sites for cellular components, such as microtubule-associated proteins (MAPs) and molecular motors[6]. However, the detailed mechanisms of how axonemal polyglutamylation regulates the stability and functionality of cilia remain to be understood.

Polyglutamylation is reversible, and tightly regulated by a balance between opposing enzymes for glutamylation or deglutamylation[7, 8]. More specifically, tubulin glutamylation is conducted by a family of tubulin tyrosine ligase-like (TTLL) proteins, including TTLL1, 4, 5, 6, 7, 9, 11, and 13[9, 10]. Each TTLL has a priority for initiation or elongation of glutamylation, as well as substrate preference between α-tubulins and β-tubulins[10]. This TTLL-mediated polyglutamylation is counteracted by a family of cytosolic carboxypeptidases (CCPs). Thus far, CCP1, 2, 3, 4, 5, and 6 have been identified as deglutamylases[6, 11]. CCP5 preferentially removes a glutamate at the branching fork, whereas other CCP members target a glutamate residue in a linear, tandem sequence in vivo[12, 13]. In contrast, Berezniuk et al. recently performed a biochemical assay to demonstrate that CCP5 cleaves glutamates at both locations and could complete the deglutamylation without the need for other CCP members[14].

The effects of tubulin polyglutamylation on the structure and functions of microtubules have been studied mainly through the following approaches: (1) biochemical characterization of gluta-mylated microtubules, (2) cell biology assays for hyperglutamy-lation or hypoglutamylation induced by genetically controlling the expression level of corresponding PTM enzymes, and (3) cell biological analysis of genetically mutated tubulins. As a result, it has been shown that chemical conjugation of glutamate side chains on purified microtubules increases the processivity and velocity of kinesin-2 motors[15]. Tubulin hyperglutamylation leads to microtubule disassembly owing to the binding of a severing enzyme, namely spastin, to hyperglutamylated microtubules[16, 17]. Mice lacking a subunit of the polyglutamylase complex display hypoglutamylation in neuronal cells, which is accompanied by a decreased binding affinity of kinesin-3 motors to microtubules[18]. Moreover, the genetic or morpholino-mediated perturbation of polyglutamylases or deglutamylases across different model organisms results in morphological and/or functional defects in cilia and flagella[19–33]. Collectively, these studies strongly suggest the importance of tubulin polyglutamylation in the structural integrity and functionality of microtubules in cilia, as well as other subcellular compartments. However, these approaches also revealed technical limitations. First, the distribution pattern of polyglutamylated tubulin is spatiotemporally dynamic; i.e., polyglutamylation is abundant in axoneme, centrioles, and neu-ronal axons in quiescent cells, which converges to the mitotic spindle and midbody during cytokinesis[6]. This dynamic feature cannot be directly addressed by conventional genetic manipula-tions or pharmacological inhibitors. Second, constitutive genetic perturbation often allows for compensation where cells adapt to their new genetic environment, likely leading to a missed detection of immediate consequences of loss-of-function, such as an effect on transient interactions between tubulins with specific

PTMs and their molecular partners[34]. Third, besides tubulin, many nucleocytoplasmic shuttling proteins such as nucleosome assembly proteins have also been identified as substrates of glutamylases[10, 35]. Therefore, global manipulation of genes encoding enzymes that modulate glutamylation is insufficient to specifically perturb axonemal glutamylation.

To circumvent these limitations, we developed a method named STRIP for SpatioTemporal Rewriting of Intraciliary PTMs. The method is based on chemically inducible dimerization (CID) to spatiotemporally recruit a catalytic domain of deglutamylase onto ciliary axoneme, where the axonemal polyglutamylation can be rapidly stripped in an inducible manner. By implementing STRIP with simultaneous live cell, time-lapse fluorescence imaging, we report cell biological analysis on the immediate consequences of deglutamylation that is rapidly induced inside primary cilia.

## Results

**Rapid relocalization of soluble proteins to the axoneme**. We begin by describing a new approach, STRIP, for the spatio-temporal perturbation of axonemal polyglutamylation to study its immediate effect on the interplay among the microtubules, molecular motors, and MAPs (Fig. 1a). The basis of our technology is chemically inducible dimerization[36, 37], in which a chemical such as rapamycin induces the dimerization of FK506 binding protein (FKBP) and FKBP–rapamycin-binding domain (FRB) (Fig. 1a). To anchor a fusion protein of Cerulean3 and FRB onto ciliary axoneme, we employed microtubule-associated protein 4 (MAP4) that accumulates at the axoneme[38]. In particular, we used the truncated mutant of MAP4m containing partial proline-rich domain and affinity domain[38] to minimize the unexpected, yet possible biological effects arising from the N-terminal region (Supplementary Fig. 1a). When expressed in ciliated cells, Cerulean3–FRB–MAP4m mainly localized inside the cilia and displayed weak signals in the cytosol (Supplementary Fig. 1b). Its ciliary localization was further confirmed by not only overlapping distribution with a ciliary membrane marker, Arl13B, but also colocalization with two axoneme markers, acetylated tubulin and glutamylated tubulin (Supplementary Fig. 1b), assuring the intended localization of the MAP4m fusion protein. We then assessed the potential effect of overexpression of Cerulean3–FRB–MAP4m on the primary cilia. First, the expres-sion of Cerulean3–FRB–MAP4m was confirmed to exhibit no significant effect on cilium length, axonemal acetylation, or axonemal glutamylation (Supplementary Fig. 1c). Next, we tested whether Cerulean3–FRB–MAP4m would affect IFT by monitoring Neon-IFT88, which labels IFT particles, with live cell, time-lapse fluorescence microscopy[39]. The rates of IFT in anterograde and retrograde directions in the cilia with Cerulean3–FRB–MAP4m were not significantly different from those under control conditions (Supplementary Fig. 1d,e). Collectively, these results confirmed that Cerulean3–FRB–MAP4m is a valid fusion construct to be anchored at the ciliary axoneme.

Our previous study showing that most cytosolic proteins have access to the ciliary lumen[40] suggests that cytosolic FKBP proteins can be trapped at the axoneme upon rapamycin addition in cells expressing Cerulean3–FRB–MAP4m. To test this possibility, we transfected NIH3T3 cells with Cerulean3–FRB–MAP4m (axoneme), 5HT$_6$–mCherry (cilia membrane), and YFP-tagged FKBP (YFP–FKBP; cytosol). Subsequent exposure to rapamycin increased the YFP fluorescence signal at the ciliary axoneme over 5 min (Fig. 1b–d; Supplementary Movie 1). The time required for half-maximal accumulation of YFP–FKBP in the axoneme ($t_{1/2}$) was 98.1 ± 24.0 s (mean ± s.e.m.; Fig. 1d). To generalize the

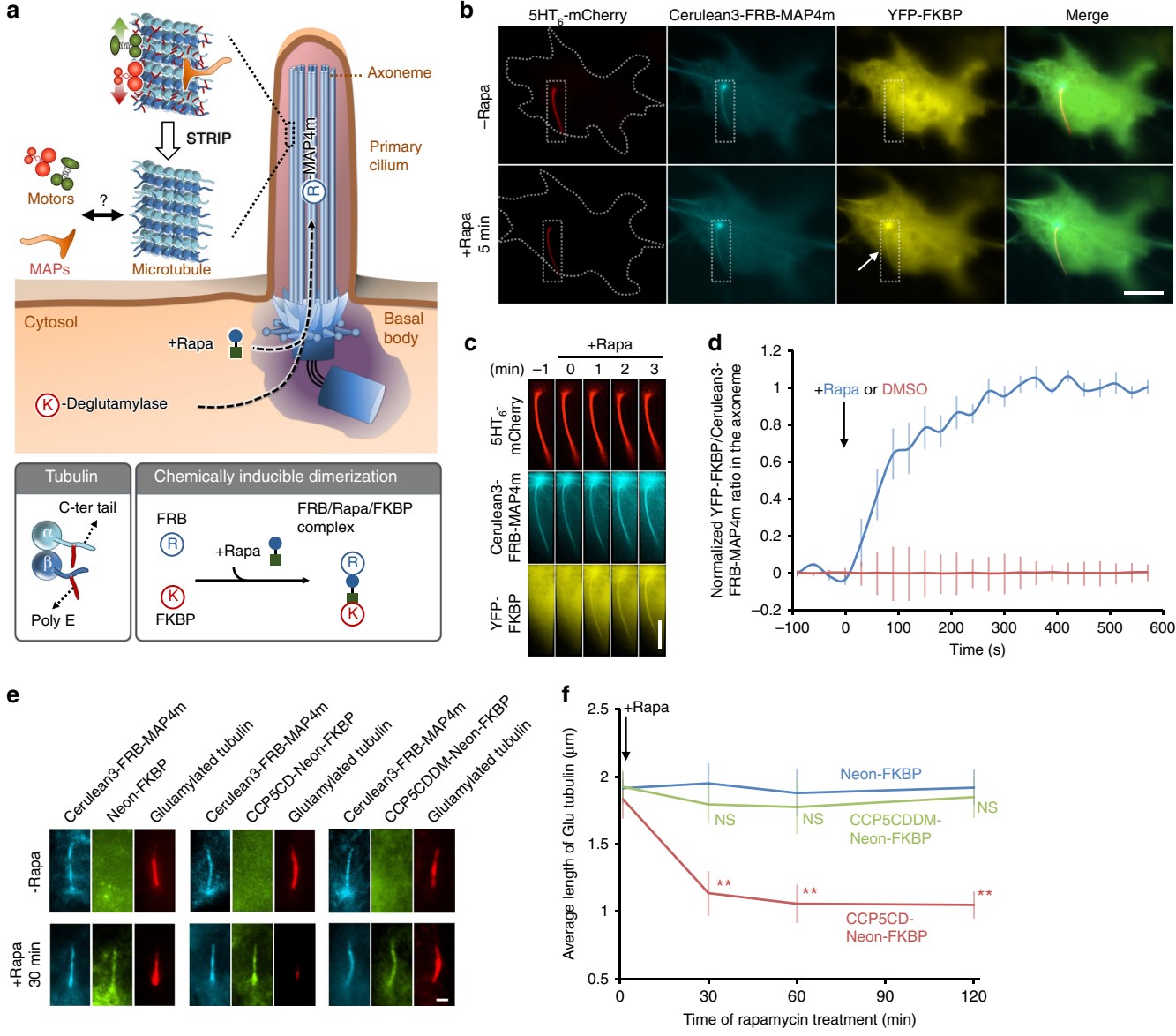

**Fig. 1** Spatiotemporally depleting tubulin glutamylation inside the cilia. **a** Schematic diagram of the STRIP (SpatioTemporal Rewriting of Intraciliary PTMs) approach which translocates an engineered deglutamylase from the cytosol onto ciliary axoneme using the chemically induced dimerization system. The cytosolic FKBP–deglutamylase can be recruited to the FRB–MAP4m-labeled axoneme by adding rapamycin (+Rapa). The causal relationship among the axonemal glutamylation, motors, and MAPs (microtubule-associated proteins) will be evaluated. **b** The addition of 100 nM rapamycin induces the accumulation of YFP–FKBP onto the Cerulean3–FRB–MAP4m-labeled axoneme (arrow) in NIH3T3 cells. Transfected cells at 80–90% confluency were serum-starved for 24 h and then incubated with 100 nM rapamycin for the indicated times. Scale bar, 10 μm. **c** Individual video frames in the axoneme region of cells in **b** upon 100 nM rapamycin treatment. Scale bar, 5 μm. Also, see Supplementary Movie 1. **d** Time course of YFP fluorescence intensity in the axoneme of NIH3T3 cells treated with 100 nM rapamycin (blue) or 0.1% DMSO (red). Data represent the mean ± s.e.m. ($n = 10$ cells for DMSO group, $n = 8$ cells for rapamycin group; four independent experiments). **e, f** Translocation of CCP5CD-Neon-FKBP to the axoneme reduces axonemal glutamylation. NIH3T3 cells were transfected with P2A-based constructs for co-expression of Cerulean3–FRB–MAP4m and Neon-FKBP-tagged proteins. Transfected cells at 80–90% confluency were serum-starved for 24 h and then incubated with 100 nM rapamycin for the indicated times. The level of axonemal glutamylation in cells expressing the indicated proteins was assessed by labeling with anti-glutamylated tubulin antibody and quantifying before and after the addition of 100 nM rapamycin. Scale bar, 1 μm. Data represent the mean ± s.e.m. ($n = 276$, 299, and 244 cells for the Neon-FKBP, CCP5CD-Neon-FKBP, and CCP5CDDM-Neon-FKBP groups, respectively; 3–5 independent experiments). NS and ** indicate no significant difference and $P < 0.01$, respectively, between the control (Neon-FKBP) and the indicated groups (Student's $t$-test)

method, we next tested the MAP4m approach with a gibberellin-based CID that works orthogonally to the rapamycin-based CID[41]. Here, we constructed two fusion proteins: a codon-optimized gibberellin insensitivity DWARF1 (mGID1) fused to neon (Neon-mGID1), and MAP4m fused to an N-terminal 92 amino acids of gibberellin insensitive (GAIs–CFP–MAP4m). The addition of a gibberellin analog (GA₃-AM) to cells coexpressing

these fusion proteins led to accumulation of the neon fluorescence signal at the ciliary axoneme (Supplementary Fig. 2; Supplementary Movie 2), albeit with slower kinetics (127.3 ± 25.9 s) compared with the rapamycin-based CID (Supplementary Fig. 2c). Taken together, these results confirmed that the CID systems utilizing MAP4m can relocate cytosolic proteins onto the ciliary axoneme of living cells within minutes.

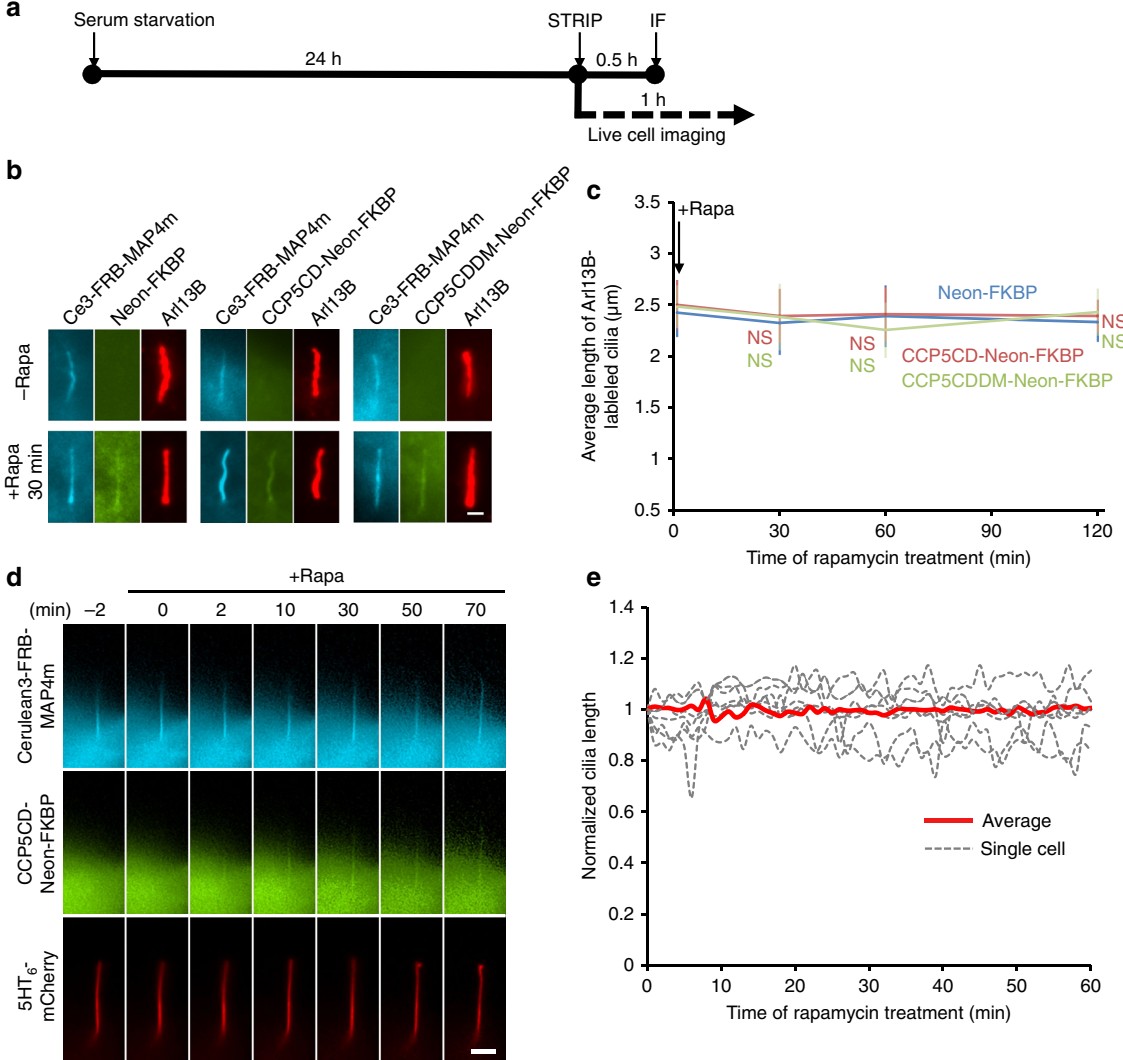

**Fig. 2** Glutamylation is not required for cilia maintenance. **a** The experimental procedure for applying STRIP together with immunofluorescence staining (IF) in **b** or live-cell imaging in **d** after serum starvation-induced ciliogenesis. **b,c** Rapid deglutamylation does not affect ciliary length in the steady state. NIH3T3 cells were transfected with P2A-based constructs for co-expression of Cerulean3 (Ce3)–FRB–MAP4m and Neon-FKBP-tagged proteins. Transfected cells at 80–90% confluency were serum-starved for 24 h and then incubated with 100 nM rapamycin for the indicated times. The cilium length in cells expressing the indicated proteins was measured by labeling with anti-Arl13B antibody and quantifying before and after the addition of 100 nM rapamycin. Scale bar, 1 μm. Data represent the mean ± s.e.m. (n = 269, 282, and 238 cells for the Neon-FKBP, CCP5CD-Neon-FKBP, and CCP5CDDM-Neon-FKBP groups, respectively; 3–5 independent experiments). NS indicates no significant difference between the control (Neon-FKBP) and the indicated groups (Student's t-test). **d** The real-time morphology of the primary cilium during axonemal deglutamylation. NIH3T3 cells were cotransfected with CCP5CD-Neon-FKBP–P2A–Cerulean3–FRB–MAP4m and a ciliary membrane marker, 5HT$_6$-mCherry. Transfected cells at 80–90% confluency were serum-starved for 24 h and then incubated with 100 nM rapamycin for the indicated times. Video frames of a cell coexpressing the indicated proteins upon treatment with 100 nM rapamycin are shown. Scale bar, 4 μm. Also see Supplementary Movie 4. **e** The length of 5HT$_6$-mCherry-labeled cilia in **d** was normalized by dividing the measured length by the length measured before rapamycin treatment and plotted. Red curves and gray dot curves show the average length and the length of each cilium during deglutamylation, respectively (n = 7 cilia from four independent experiments)

**Relocalizing CCP5 to the axoneme for rapid deglutamylation.**
To change the polyglutamylation status of tubulins in cilia, we implemented STRIP to recruit a deglutamylase to the axoneme. Among the tubulin deglutamylase family, CCP5 is demonstrated to efficiently remove glutamate residues at a branch point, as well as those in a linear tandem both in vitro and in vivo[12, 14], suggesting CCP5 as a valid candidate to actuate deglutamylation upon the STRIP execution. We thus constructed Neon-FKBP-tagged full-length CCP5 (CCP5FL-Neon-FKBP), intending to relocalizing this fusion protein from the cytosol to the axoneme upon rapamycin addition. However, CCP5FL-Neon-FKBP localized inside the cilia even prior to rapamycin addition,

thereby constitutively depleting axonemal glutamylation (Supplementary Fig. 3a–c), clearly hampering the use of the full-length CCP5 for the STRIP operation. Therefore, we assessed the subcellular localization of truncated mutants of CCP5, namely N-terminus (CCP5N, residues 1–160) and the catalytic domain (CCP5CD, residues 161–531). When expressed in cells as a fusion protein of Neon-FKBP, these two CCP5 truncations localize in the cytosol (Supplementary Fig. 3b). For further characterization of these truncated mutants, we introduced two point mutations (H252S and E255Q) to CCP5CD (CCP5CDDM) to impair the deglutamylation activity[9]. A fusion construct of this mutant (CCP5CDDM-Neon-FKBP) was also confirmed to be cytosolic

(Supplementary Fig. 3b). In contrast to full-length CCP5, none of the truncated mutants had an impact on axonemal glutamylation (Supplementary Fig. 3b,c). To directly assess the enzymatic activities of these CCP5 truncations, we forcefully anchored each of them to the axoneme via fusion with MAP4m (Supplementary Fig. 4a). The immunofluorescence assay shows that CCP5CD, but not CCP5N or CCP5CDDM, significantly reduces axonemal glutamylation, indicating that CCP5CD retains viable enzyme activity (Supplementary Fig. 4b,c). Importantly, axonemal deglutamylation induced by CCP5FL-Neon-FKBP or CCP5CD–Cerulean3–MAP4m did not affect axonemal acetylation or cilium length (Supplementary Figs. 3c and 4c). Taken together, our results identified CCP5CD as an ideal candidate that becomes functional only at the axoneme without affecting other aspects of primary cilia such as cilia length and tubulin acetylation status.

To trigger deglutamylation of axonemal tubulin, CCP5CD-Neon-FKBP was recruited to the axoneme with Cerulean3–FRB–MAP4m by adding rapamycin (Supplementary Fig. 5 and Supplementary Movie 3). We inserted a viral P2A self-cleaving linker in-between FKBP and FRB pieces, which encodes relatively equal amounts of two proteins in subsequent experiments[42]. The kinetics of CCP5CD-Neon-FKBP accumulation in the axoneme upon rapamycin treatment were comparable to those of YFP–FKBP (CCP5CD-Neon-FKBP: $95.7 \pm 21.9$ s; YFP–FKBP: $98.1 \pm 24.0$ s). The recruitment of CCP5CD-Neon-FKBP to the axoneme reduced the glutamylation within 30 min (Fig. 1e, f). Interestingly, deglutamylation did not alter the localization of MAP4m, suggesting that MAP4m binding to microtubules is independent of tubulin glutamylation (Fig. 1e). Importantly, the recruitment of Neon-FKBP alone or catalytically inactive CCP5CDDM onto the axoneme was insufficient to deplete axonemal glutamylation, confirming that CCP5CD-induced axonemal deglutamylation is dependent on the enzymatic activity of CCP5 (Fig. 1e, f). It is noteworthy that the CCP5-induced or STRIP-induced deglutamylation left residual

glutamylation at the proximal end of primary cilia (Fig. 1e; Supplementary Fig. 3b). The residual glutamylation localized at the inversin zone instead of the transition zone (Supplementary Fig. 6). The detailed mechanism of how axonemal tubulins in the inversin zone resist deglutamylation treatment is unclear.

**Specificity of CCP5–STRIP in quality and space.** The status of polyglutamylation was verified by two additional antibodies, anti-polyE and anti-Δ2-tubulin antibodies, that recognize long glutamate side chains and penultimate glutamate in the primary sequence of tubulin, respectively[12]. The recruitment of CCP5CD onto the axoneme does not significantly alter the level of long glutamate side chains (Supplementary Fig. 7a, b) or Δ2 tubulin (Supplementary Fig. 7c, d), suggesting the insufficient activity of CCP5 in shortening polyglutamate side chains or catalyzing the formation of Δ2–Δ3 tubulins, respectively (Supplementary Fig. 7a–d).

To determine the spatial specificity of STRIP-triggered axonemal deglutamylation, we measured the level of tubulin glutamylation in other subcellular compartments. Glutamylated tubulins in the cytosol and basal body can be visualized after "cold treatment" before fixation (see the Methods section, Supplementary Fig. 8a)[10]. Consistent with previous reports[8, 12], deglutamylation by CCP5FL is prevalent in the cytosol, basal body, and axoneme (Supplementary Fig. 8a,b). In contrast, CCP5CD does not lead to a noticeable change in tubulin glutamylation compared with a control condition with neon alone. Strikingly, the STRIP-triggered deglutamylation was most prominent at the axoneme, and nonsignificant elsewhere in the same single cells, validating the high spatial precision (Supplementary Fig. 8a,b). It is somewhat interesting that the expression of soluble CCP5CD in cells had only a marginal effect on glutamylation in the cytosol, despite the nature of free diffusion. We speculate that the expression level of cytosolic CCP5CD may be low enough to take an effect, and/or that

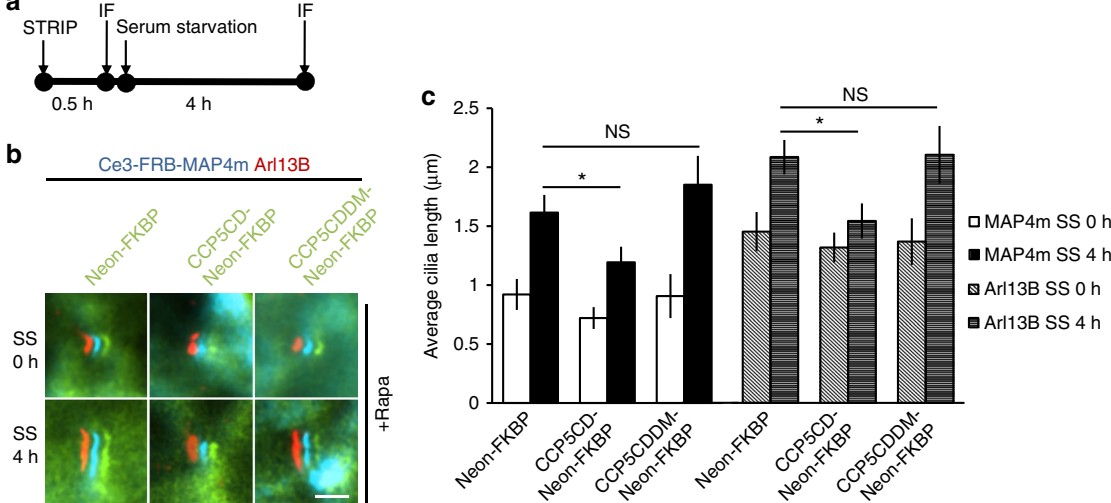

**Fig. 3** Glutamylation is important for cilia elongation. **a** Experimental procedure for applying STRIP together with immunofluorescence staining (IF) in **b** before and after serum starvation-induced cilia elongation. Washout of rapamycin after a 30-min incubation did not affect protein dimerization in cells owing to the irreversible nature of the CID system. **b,c** Rapid deglutamylation inhibits serum starvation-induced cilia elongation. NIH3T3 cells were transfected with P2A-based constructs for co-expression of Cerulean3–FRB–MAP4m and Neon-FKBP-tagged proteins. Transfected cells at 80–90% confluency were incubated with 100 nM rapamycin for 30 min and then serum-starved for the indicated times. The length of the axoneme and of cilia was quantified in cells expressing the indicated proteins with rapamycin treatment before and after serum starvation (SS) for 4 h. The images show shifted overlays of the indicated proteins in cilia. Scale bar, 2 µm. **c** Average length of cilia labeled by MAP4m and Arl13B in **b**. Data represent the mean ± s.e.m. (*n* = 83, 80, 58, 59, 102, 92, 65, 82, 52, 51, 53, and 51 cells from left to right; three independent experiments). NS and * indicate no significant difference and *P* < 0.05, respectively, between the control (Neon-FKBP) and the indicated groups (Student's *t*-test)

endogenous polyglutamylases in the cytosol may counteract CCP5CD-mediated deglutamylation in the cytosol. Nevertheless, a design principle of the STRIP enables drastic concentration of CCP5CD on the axoneme at the expense of a fraction of the CCP5CD pool in the cytosol, notably due to >10,000-fold volume difference between cilia and the cytosol.

**The roles of glutamylation in the ciliary structure**. To reveal the effect of axonemal deglutamylation on the ciliary structure, we first measured the cilia length based on Arl13B staining. The cilium length measured before and after STRIP treatment at least for 2 h was comparable among the control (no CCP5), wild-type (CCP5CD), and inactive mutant (CCP5CDDM) (Fig. 2b, c). Next, we assessed the length and morphology of primary cilia in live cells. Cells expressing CCP5CD-Neon-FKBP, Cerulean3–FRB–MAP4m, and 5HT$_6$–mCherry underwent live-cell fluorescence imaging. Rapamycin addition does not result in alteration of the cilium length or morphology (Fig. 2d, e; Supplementary Movie 4). As expected from the result with CCP5CD–Cerulean3–MAP4m, STRIP-mediated deglutamylation

does not affect axonemal detyrosination and acetylation (Supplementary Figs. 7e, f, 9). These results indicate that the structural integrity of steady-state primary cilia is organized independently of glutamylation.

Axoneme growth takes place in an early stage of ciliogenesis, and associates with polyglutamylation[43, 44]. To study the role of polyglutamylation in the growing axoneme, we rapidly degluta-mylated axoneme under a serum-starved condition to induce cilia elongation (Fig. 3). MAP4m accumulates with no problem at the growing axoneme, enabling the STRIP operation for deglutamyla-tion (Supplementary Fig. 10 and Supplementary Movie 5). Recruitment of CCP5CD onto the growing axoneme significantly slows down the extension of cilium length (Fig. 3b,c). Taken together, these results indicate that axonemal glutamylation is important for cilia elongation during ciliogenesis, but not for maintenance of steady-state cilia.

**Deglutamylation hampers kinesin-2-mediated anterograde IFT**. Besides mechanical support, the axoneme also serves as railways for IFT[45, 46]. We thus evaluated whether axonemal

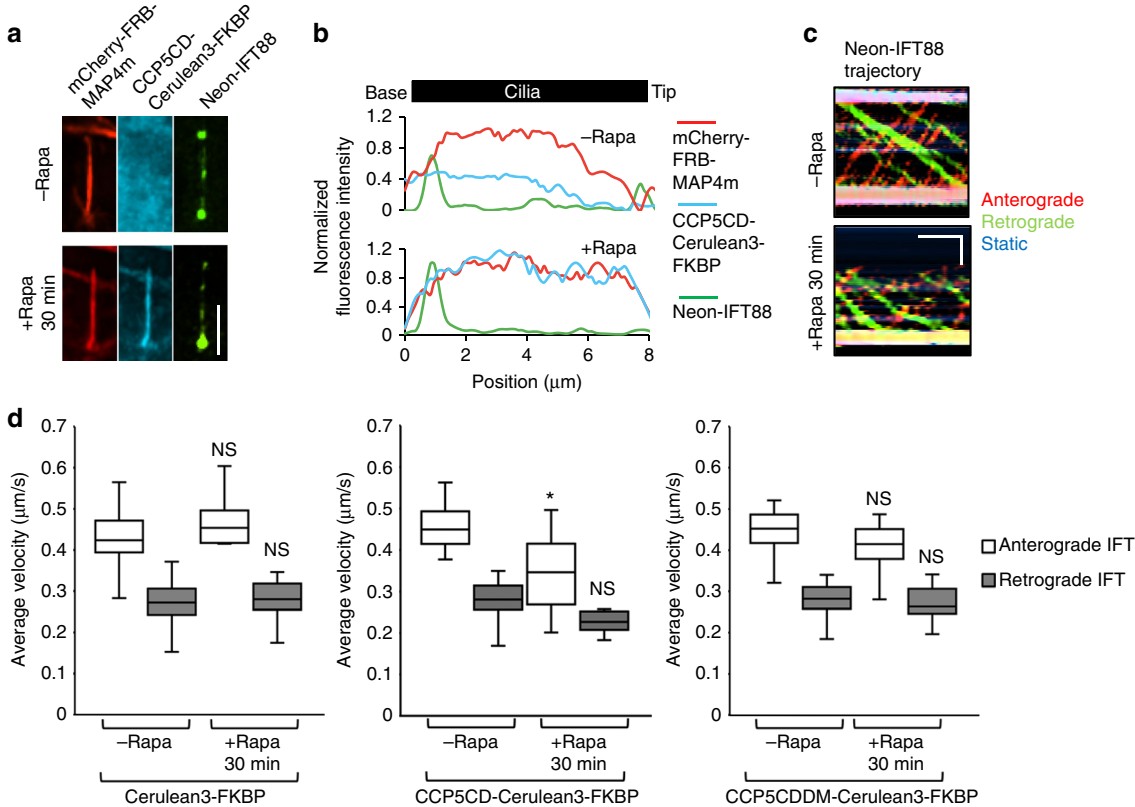

**Fig. 4** Rapid axonemal deglutamylation preferentially hampers anterograde IFT. **a** Neon-IFT88 accumulates at the base of deglutamylated cilia. Neon-IFT88-stable NIH3T3 cells were transfected with CCP5CD-Cerulean3-FKBP-P2A–mCherry–FRB–MAP4m. Transfected cells at 80–90% confluency were serum-starved for 24 h and then treated with 100 nM rapamycin for 30 min. Scale bar, 4 μm. **b** Linescan profile of the indicated proteins from base to the tip of the primary cilium in **a**. **c** Representative kymographs of Neon-IFT88 generated from time-lapse imaging of the cilium before and after rapamycin treatment for 30 min. Red, green, and blue lines represent the trajectories of Neon-IFT88 particles in anterograde and retrograde directions, and static Neon-IFT88, respectively. Horizontal scale bar, 10 s. Vertical scale bar, 2 μm. Also, see Supplementary Movie 6. **d** Translocation of CCP5CD–Cerulean3–FKBP, but not Cerulean3–FKBP or CCP5CDDM–Cerulean3–FKBP, onto the axoneme hampers the IFT only in the anterograde direction. Neon-IFT88-stable NIH3T3 cells were transfected with P2A-based constructs for co-expression of mCherry–FRB–MAP4m and Cerulean3–FKBP-tagged proteins. Transfected cells at 80–90% confluency were serum-starved for 24 h and then treated with 100 nM rapamycin for the indicated times. The velocity of Neon-IFT88 was quantified according to the trajectories shown in the kymographs (see the Methods section; $n = 175$, 150, and 170 Neon-IFT88 particles for the Cerulean3–FKBP, CCP5CD–Cerulean3–FKBP, and CCP5CDDM–Cerulean3–FKBP groups, respectively; three independent experiments). NS and * indicate no significant difference and $P < 0.05$, respectively, between the conditions in the presence or absence of rapamycin (Student's $t$-test)

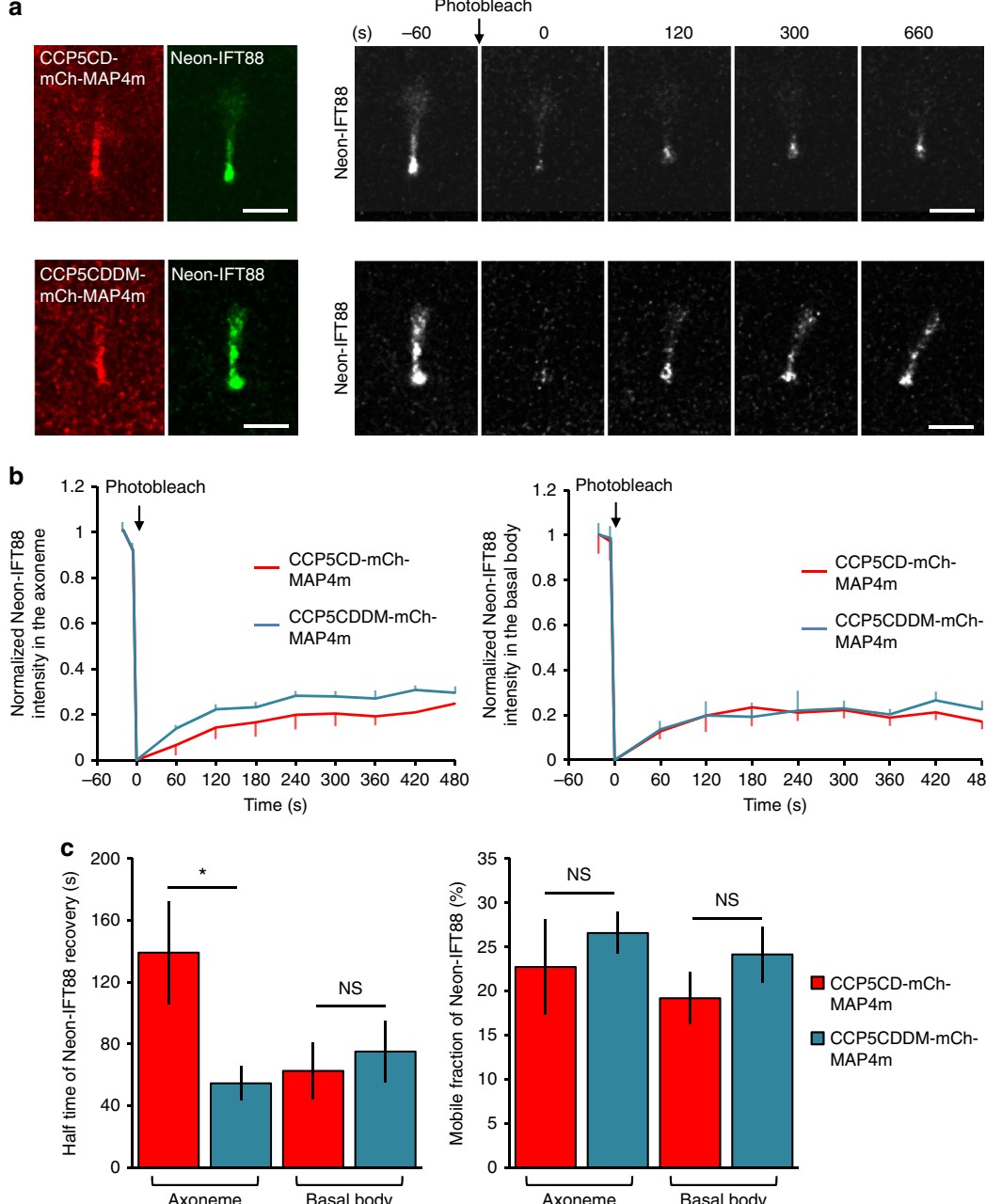

**Fig. 5** Axonemal deglutamylation hampers IFT dynamics along the axoneme but not the tethering of IFT machinery on the basal body. **a** Neon-IFT88-stable NIH3T3 cells were transfected with CCP5CD–mCherry–MAP4m or catalytically inactive CCP5CDDM–mCherry–MAP4m. Transfected cells at 80–90% confluency were serum-starved for 24 h prior to FRAP experiments. Cells expressing the indicated proteins were photobleached in the entire cilia region and allowed to recover for the indicated times. Scale bar, 2 μm. **b** Fluorescence recovery of Neon-IFT88 in the axoneme region (left) and basal body (right) was measured and plotted. **c** The recovery rate (left) and mobile fraction (right) of Neon-IFT88 in the axoneme and basal body were measured and plotted. Data represent the mean ± s.e.m. ($n = 9$ and 13 cilia for the CCP5CD–mCherry–MAP4m and CCP5CDDM–mCherry–MAP4m groups, respectively; four independent experiments). NS and * indicate no significant difference and $P < 0.05$, respectively, between the CCP5CD-mCherry-MAP4m and CCP5CDDM-mCherry-MAP4m groups (Student's $t$-test)

deglutamylation impacts the rate of IFT. To test this, the IFT activity was quantified before and after STRIP-mediated axonemal deglutamylation by monitoring Neon-IFT88 with live-cell fluorescence microscopy (Fig. 4). A subsequent linescan analysis of the Neon-IFT88 signal in individual cilia at two time points (before and 30 min after rapamycin treatment) showed that Neon-IFT88 initially spread across the entire cilia in the form of puncta converges to the base of the cilium after deglutamylation (Fig. 4a, b). To assess IFT motility, a kymograph was then generated based on the time-lapse images of Neon-IFT88 before and after deglutamylation (Fig. 4c). The velocity of IFT in both directions (i.e., anterograde and retrograde) for control and deglutamylated cilia was calculated based on the slopes of each IFT88 trajectory on the kymograph (Fig. 4c). The rates of bidirectional IFT were comparable in both control groups (Fig. 4d, left and right panels). In contrast, anterograde but not retrograde IFT was significantly slow after axonemal deglutamylation by CCD5CD ($0.46 \pm 0.03$ μm/s vs. $0.37 \pm 0.03$ μm/s) (Fig. 3d, middle panel; Supplementary Movie 6). These results demonstrated that rapid deglutamylation preferentially affects anterograde IFT.

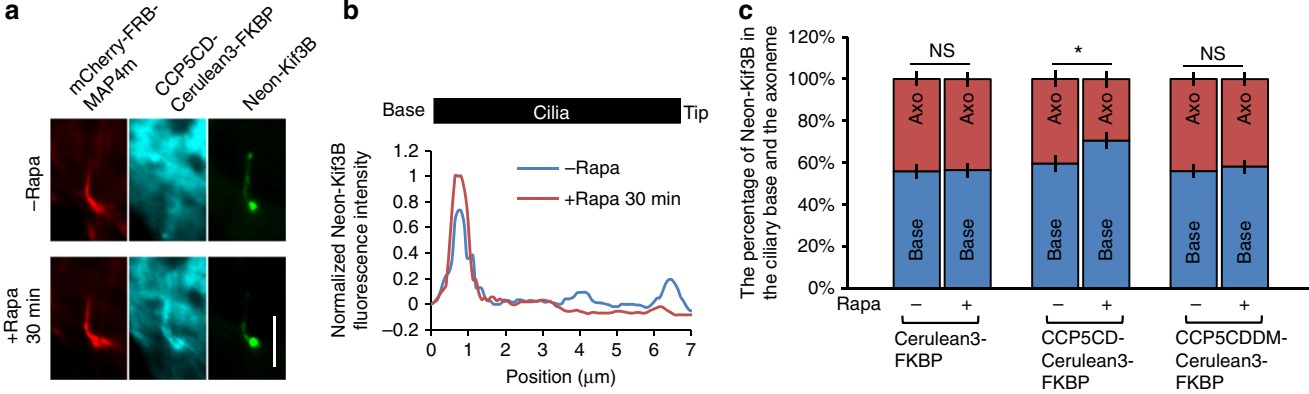

**Fig. 6** Kinesin-2 accumulates at the base of deglutamylated cilia. **a** Neon-Kif3B-stable NIH3T3 cells were transfected with CCP5CD–Cerulean3–FKBP–P2A–mCherry–FRB–MAP4m. Transfected cells at 80–90% confluency were serum-starved for 24 h and then treated with 100 nM rapamycin for 30 min. Scale bar, 4 μm. **b** Linescan profile of Neon-Kif3B from the base to the tip of the cilium in **a**. **c** Neon-Kif3B-stable NIH3T3 cells were transfected with P2A-based constructs for co-expression of mCherry–FRB–MAP4m and Cerulean3–FKBP-tagged proteins. Transfected cells at 80–90% confluency were serum-starved for 24 h and then treated with or without 100 nM rapamycin for 30 min. The percentage of Neon-Kif3B at the base (Base) or axoneme region (Axo) of control or deglutamylated cilia is plotted. Data represent the mean ± s.e.m. (n = 74, 48, and 69 cells for the Cerulean3–FKBP, CCP5CD-Cerulean3–FKBP, and CCP5CDDM–Cerulean3–FKBP groups, respectively; three independent experiments). NS and * indicate no significant difference and P < 0.05, respectively, between the conditions in the presence and absence of rapamycin (Student's t-test)

Besides rapid deglutamylation, we also evaluated the effect of long-term deglutamylation on IFT dynamics by measuring the velocity of Neon-IFT88 in cilia expressing CCP5CD–mCherry–MAP4m (Supplementary Fig. 11). Cilia expressing catalytically inactive CCP5CDDM–mCherry–MAP4m m were included as a control (Supplementary Fig. 11). Consistent with the finding that rapid deglutamylation attenuated anterograde IFT, long-term deglutamylation induced by CCP5CD–mCherry–MAP4m significantly slowed down anterograde IFT but not retrograde IFT (Supplementary Fig. 11b). Moreover, long-term deglutamylation only slightly but not significantly decreased the frequency of Neon-IFT88 in both directions (Supplementary Fig. 11c).

We next used fluorescence recovery after photobleaching (FRAP) to measure the ciliary entry of IFT in control and deglutamylated cilia (Fig. 5). Neon-IFT88 in the whole ciliary region was photobleached, and the fluorescence recovery at the basal body or axoneme was measured. The recovery rate and mobile fraction of Neon-IFT88 at the ciliary base were compared between control and deglutamylated cilia (Fig. 5; Supplementary Movie 7), which indicated that axonemal deglutamylation does not affect the tethering of Neon-IFT88 to the basal body. Interestingly, axonemal deglutamylation induced by CCP5CD–mCherry–MAP4m significantly slowed down the recovery rate of Neon-IFT88 onto the axoneme ($t_{1/2}$ in control cilia: 54.43 ± 11.24 s; $t_{1/2}$ in deglutamylated cilia: 139.14 ± 33.49 s), probably owing to defects in anterograde IFT induced by axonemal deglutamylation (Fig. 5; Supplementary Movie 7). Taken together, these results confirmed that axonemal deglutamylation slows down anterograde IFT without blocking the tethering of the IFT machinery to the basal body.

Anterograde IFT is mainly powered by kinesin motors[45], whose distribution and motility are reportedly modulated by tubulin glutamylation[15, 18]. We therefore hypothesized that axonemal deglutamylation inhibits anterograde IFT by impairing kinesin motility. To address this, we examined the effects of axonemal deglutamylation on the distribution of Neon-Kif3B, a motor subunit of the kinesin-2 complex. Neon-Kif3B was observed to move along the axoneme (Supplementary Fig. 12a,b and Supplementary Movie 8). The anterograde rate of Neon-Kif3B particles was measured to be comparable to that of the

Neon-IFT88 trains (Supplementary Fig. 12b,c), suggesting that Neon-Kif3B can be used to assess the motility of kinesin motors. We then found that Neon-Kif3B accumulates at the proximal end of deglutamylated cilia (Fig. 6a, b), just like Neon-IFT88 (Fig. 4a, b). To reinforce this finding, we performed the quantification of Neon-Kif3B distribution in the cilia divided into two compartments, base and axoneme, which exhibited more Neon-Kif3B at the base of deglutamylated cilia compared with control cilia (70.7 ± 2.2% vs. 59.6 ± 3.9%) (Fig. 6c). In summary, these results suggest that glutamylation controls anterograde IFT through Kif3B of the kinesin-2 complex.

The Lechtreck group once showed that the demand of anterograde IFT in newly growing cilia is much higher than that in mature cilia[47]. Therefore, the rapid deglutamylation-induced defects in anterograde IFT may cause more severe inhibition of cilia elongation than cilia maintenance, a prediction consistent with our finding that rapid deglutamylation impaired the elongation of immature, but not mature cilia (Figs. 2 and 3).

**Axonemal deglutamylation inhibits Hh signaling.** We next investigated whether axonemal deglutamylation affects ciliary Hh signaling. The Hh receptor Patched and the orphan G-protein-coupled receptor (Gpr161) localize in cilia and block Hh signaling in the absence of Hh ligands. These two signaling molecules exit from the cilium once the Hh pathway is activated[48]. Immunofluorescence assay revealed that STRIP-induced axonemal deglutamylation did not affect the ciliary exit of Patched1-YFP and GPR161 upon the treatment of Smo agonist, SAG (Supplementary Fig. 13), indicating that glutamylation is not involved in modulating these two upstream signaling components in Hh signaling.

The kinesin-2 complex interacts with Hh components to regulate Hh signaling[48–52], leading us to hypothesize that deglutamylation-induced defects in kinesin-2-mediated anterograde IFT would collaterally affect Hh signaling. We thus examined the subcellular location of Hh components such as Smoothened (Smo) and Gli3 in control and deglutamylated cilia after stimulation with the SAG (Fig. 7a). The SAG-induced accumulation of Smo (Fig. 7b, c) and Gli3 (Fig. 7d, e) in the cilia was significantly impaired by axonemal deglutamylation. The

distribution and motility of GFP–Gli3 in deglutamylated cilia upon treatment with SAG was also evaluated by FRAP (Supplementary Fig. 14). Deglutamylation induced by CCP5CD–mCherry–MAP4m suppressed the ciliary distribution of GFP–Gli3 upon stimulation with SAG (Supplementary Fig. 14b), which is consistent with the results obtained with

STRIP operation (Fig. 7d, e). Moreover, axonemal deglutamylation significantly slowed down the entry of GFP–Gli3 into the cilia, probably owing to defective anterograde IFT (Supplementary Fig. 14c,d and Supplementary Movie 9).

We also tested whether axonemal deglutamylation inhibits Gli activation using 8xGBS–GFP:NIH3T3 reporter cells in which a

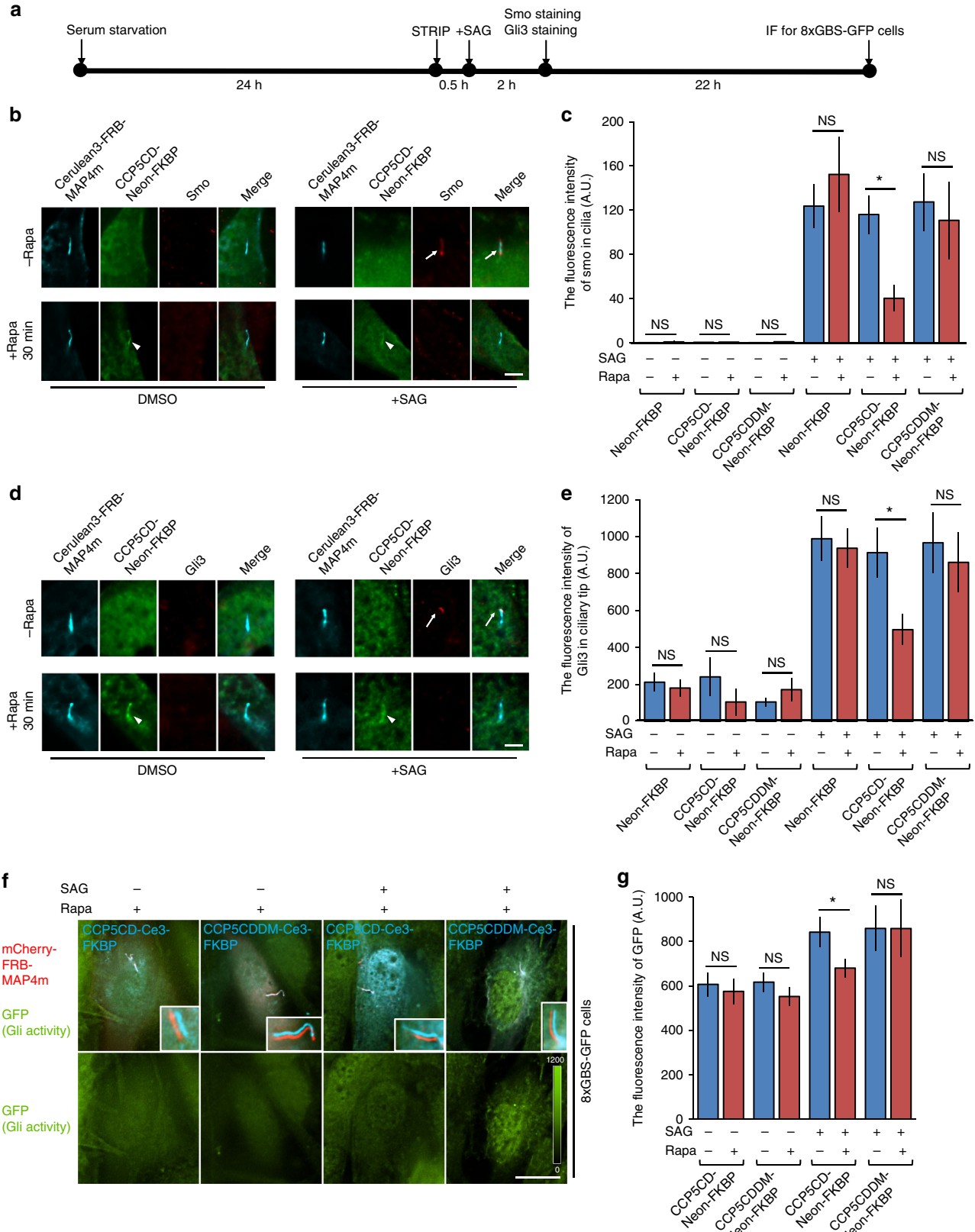

minimal promoter and 8xGli-binding site (GBS) drive green fluorescent protein (GFP) expression[53]. GFP served as a fluorescence reporter to represent Gli transcription activities. As expected, treatment with SAG significantly increased GFP intensity in these reporter cells with no STRIP operation (cells expressing CCP5CDDM–Cerulean3–FKBP; Fig. 7f, g). However, STRIP-mediated axonemal deglutamylation abolished the GFP fluorescence increase in SAG-treated cells (cells expressing CCP5CD–Cerulean3–FKBP; Fig. 7f, g). Taken together, these results confirmed that axoneme glutamylation is important for Hh signaling, likely through an anterograde IFT-dependent mechanism.

## Discussion

In this study, we developed a new approach based on the CID system for spatiotemporally perturbing the tubulin PTMs in living cells. This is achieved by the inducible dimerization between an axonemal targeting protein, FRB-tagged MAP4m, and an engineered PTM modification enzyme, CCP5CD–FKBP. Local accumulation of CCP5CD onto MAP4m-labeled axoneme depletes axonemal glutamylation within minutes, which enables us to uncover that glutamylation is involved in ciliogenesis (Figs. 3 and 8), anterograde IFT (Figs. 4–6, and 8; Supplementary Fig. 11), and Hh signaling (Figs. 7 and 8). In contrast, glutamylation played little role in other tubulin PTMs such as acetylation and detyrosination (Supplementary Figs. 7 and 9), retrograde IFT (Fig. 4; Supplementary Fig. 11), or length maintenance of mature cilia (Fig. 2; Supplementary Figs. 3, 4, 9).

Defective ciliogenesis was previously observed after complete genetic depletion or mutation of kinesin-2 in ciliated organisms[54–57], which seems at first glance inconsistent with our present finding that deglutamylation-induced defects in kinesin-2 do not affect cilia maintenance. There are at least two explanations for this apparent discrepancy. In addition to the axoneme, kinesin-2 also localizes at the ciliary base to regulate its organization[58] (Fig. 6a; Supplementary Fig. 12a). Therefore, genetic manipulation of kinesin-2 may have affected the functions of kinesin-2 not only at the axoneme but also at the ciliary base, possibly resulting in defective basal bodies that could devastate ciliogenesis. Another explanation is that residual anterograde IFT in the STRIP cells may have been sufficient for cilia growth and maintenance, unlike the cases for near-complete loss-of-function of kinesin-2 proteins[54–57].

Our work showed that axonemal deglutamylation preferentially hampers anterograde IFT but not retrograde IFT (Fig. 4c,d; Supplementary Fig. 11). In several ciliated model organisms, glutamylation is primarily abundant on the B-tubules

of the outer-axoneme doublets[43, 59–61]. Polyglutamylase and deglutamylase mutations in several studies frequently cause defects of B-tubules in the axoneme doublets[20, 21, 62], with one exception that abnormal A-tubules were observed in TTLL6 morphants[25]. Pigino and colleagues used correlative fluorescence and three-dimensional electron microscopy to elegantly demonstrate that anterograde IFT trains move along the B-tubule, while retrograde IFT uses the A-tubule in Chlamydomonas flagella[46]. Thus, it will be interesting to determine which tubule doublet (A, B, or both) causes the defective anterograde IFT after tubulin deglutamylation. In addition to mammalian primary cilia and Chlamydomonas flagella, several recent studies demonstrated that perturbation of specific tubulin isotypes and tubulin glutamylation affects ciliary ultrastructures and impacts the motility of different anterograde kinesin motors in C. elegans cilia[63, 64]. The relationship among tubulin codes, microtubule ultrastructures, and molecular motors in different ciliated model organisms merits comprehensive scrutiny.

By verifying the effect of deglutamylation on the distribution of various Hh signaling molecules, as well as Gli transcriptional activities, we claimed that axonemal glutamylation positively regulates Hh signaling presumably through anterograde IFT-dependent mechanisms (Fig. 7; Supplementary Figs. 13 and 14). Our results show that axonemal deglutamylation attenuates Hh signaling mainly by disturbing the ciliary entry of Smo and Gli proteins instead of the removal of their upstream negative regulators from cilia (Fig. 7; Supplementary Figs. 13 and 14). Previous studies found that the ciliary entry of Gli protein is driven by anterograde IFT, which offers a legitimate explanation on how deglutamylation-induced defects in anterograde IFT inhibit Hh signaling[49, 52]. However, several studies using the pulse-chase assay and single-molecule imaging have demonstrated that the ciliary entry of Smo depends on lateral diffusion rather than anterograde IFT[65, 66]. Further work is required to decipher whether kinesin-2 assists Smo in crossing the diffusion barrier at the cilia base[48, 67].

One of the advantages of the STRIP is its modular nature originating from the CID approach[37] with which CCP5 and MAP4m can be respectively replaced with other tubulin-modification enzymes and other microtubule-binding proteins[68]. Indeed, we have also applied the STRIP system to recruit the catalytic domain of TTLL6 protein onto ciliary axoneme (Supplementary Fig. 15). Moreover, by implementing both rapamycin-mediated and gibberellin-mediated STRIPs, it would be possible to rewrite two different PTMs. Moreover, our dimerization approach can be advanced to reversible operation by utilizing light-induced dimerization systems such as Cry2–CIBN and iLid–SspB[69]. Such an enhanced STRIP will enable targeting

**Fig. 7** Axonemal deglutamylation suppresses Hedgehog signaling. **a** The experimental procedure for STRIP and Hedgehog induction. Washout of rapamycin after 30 min of STRIP operation did not affect protein dimerization in cells owing to the irreversible nature of the CID system. **b** Axonemal deglutamylation reduces the level of Smoothened (Smo) in cilia after treatment with SAG. NIH3T3 cells were transfected with P2A-based constructs for co-expression of Cerulean3–FRB–MAP4m and Neon-FKBP-tagged proteins. Transfected cells at 80–90% confluency were serum-starved for 24 h and then incubated with 100 nM rapamycin or 0.1% DMSO for 30 min and subsequently treated with 1 μM SAG for 2 h. Scale bar, 5 μm. **c** Smoothened in cilia was quantified after the indicated treatments (n ≥ 18 cells from three independent experiments). **d** Axonemal deglutamylation reduces the level of Gli3 at the ciliary tip after treatment with SAG. NIH3T3 cells were transfected with P2A-based constructs for co-expression of Cerulean3–FRB–MAP4m and Neon-FKBP-tagged proteins. Transfected cells at 80–90% confluency were serum-starved for 24 h and then incubated with 100 nM rapamycin or 0.1% DMSO for 30 min and subsequently treated with 1 μM SAG for 2 h. Scale bar, 3 μm. **e** Gli3 at the ciliary tip was quantified after the indicated treatments. Data represent the mean ± s.e.m. (n ≥ 21 cells from three independent experiments). **f** Axonemal deglutamylation inhibits Gli activation upon Hedgehog stimulation. NIH3T3:8xGBS–GFP cells were transfected with P2A-based constructs for co-expression of mCherry–FRB–MAP4m and Cerulean3 (Ce3)–FKBP-tagged proteins. Transfected cells at 80–90% confluency were serum-starved for 24 h and then incubated with 100 nM rapamycin or 0.1% DMSO for 30 min and subsequently treated with 200 nM SAG for 24 h. The GFP intensity of cells was scaled to the same ranges in each image. The insets show the shifted overlays of the indicated proteins in cilia. Scale bar, 20 μm. **g** GFP intensity of NIH3T3:8xGBS–GFP was measured under the indicated conditions (n ≥ 13 cells from three independent experiments). Data represent the mean ± s.e.m. NS and * indicate no significant difference and P < 0.05, respectively, between the conditions in the presence and absence of rapamycin (Student's t-test)

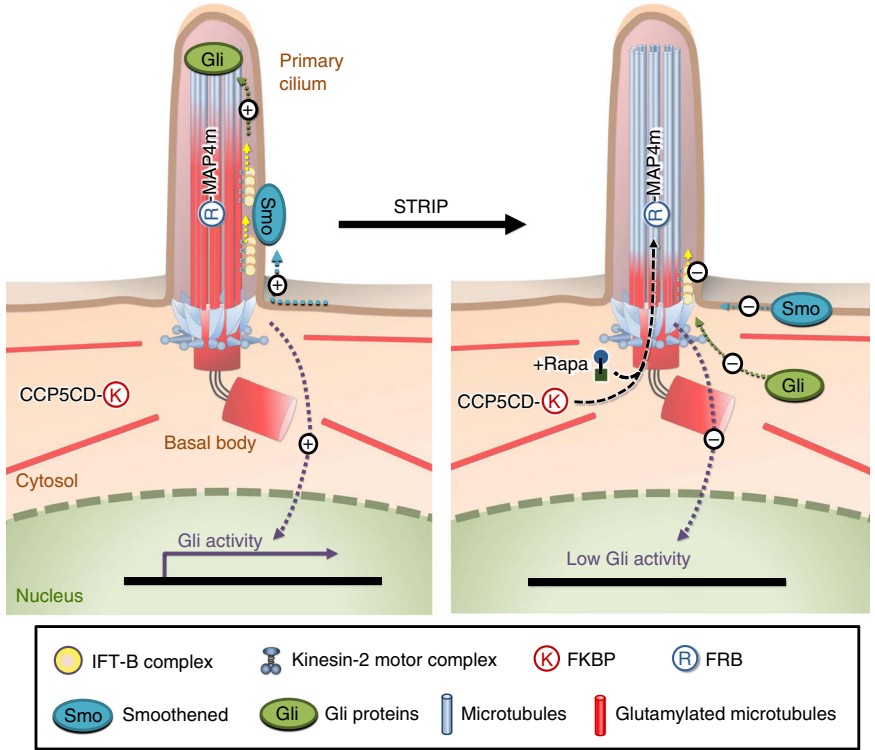

**Fig. 8** Model of our inducible axonemal deglutamylation system, STRIP, and proposed roles for tubulin glutamylation. The proximal end of the axoneme is subjected to polyglutamylation. Axonemal glutamylation facilitates kinesin-2-mediated anterograde IFT and the ciliary entry of the Hedgehog pathway components Smoothened (Smo) and Gli3 upon Hedgehog stimulation. The translocation of the CCP5 catalytic domain (CCP5CD) onto the MAP4m-appended axoneme by adding rapamycin specifically strips axonemal glutamylation and subsequently inhibits anterograde IFT and Hedgehog signaling without noticeably affecting the ciliary structure in the steady state

of a specific pair of tubulin PTMs and tubulin subtypes, which may help understand the tubulin code more thoroughly in the near future.

## Methods

**Cell culture and transfection**. NIH3T3 cells, CFP–FRB–MAP4m-expressing stable NIH3T3 cells, Neon-IFT88-expressing stable NIH3T3 cells, and Neon-Kif3B-expressing stable NIH3T3 cells were maintained at 37 °C in 5% $CO_2$ in DMEM (Corning) supplemented with 10% fetal bovine serum (Gibco), penicillin, and streptomycin (Corning). NIH3T3:8xGBS–GFP reporter line was cultured in DMEM containing 10% FBS and 2 mM GlutaMAX (GIBCO). To induce cilio-genesis, cells were serum-starved for 24 h. For induction of Hh pathway, ciliated cells were treated with 1 μM SAG (Enzo) for 2 h or 200 nM SAG for 24 h. Plasmid DNA transfection was performed by LT-1 transfection reagent (Mirus) or X-tremeGENE 9 (Roche) 24 h prior to the serum starvation. Transfected cells were treated with 100 nM rapamycin (LC Laboratories) or 100 μM $GA_3$-AM[41] for rapid induction of protein dimerization and translocation in living cells.

**Generation of NIH3T3 stable cell lines**. NIH3T3 cells stably expressing the respective proteins were generated using the MSCV retroviral expressing system. The open reading frames of the respective proteins together with the CMV promoter were subcloned into the MSCV vector (Clontech) by in-Fusion HD cloning kit (Clontech). Platinum 293T cells were transfected with MSCV–CMV–CFP–FRB–MAP4m, MSCV–CMV–Neon-IFT88, or MSCV–CMV–Neon-Kif3B for generating retrovirus particles. Retrovirus harboring the respective genes was incubated with NIH3T3 cells in the presence of 10 μg/ml polybrene (Sigma), and the infected cells were selected by 2.5 μg/ml puromycin (Sigma).

**Live-cell imaging**. Cells were plated on poly(D-lysine)-coated borosilicate glass Lab-Tek eight-well chambers (Thermo Scientific). Live-cell imaging was performed using a Nikon T1 inverted fluorescence microscope (Nikon) with a ×60 oil objective (Nikon), DS-Qi2 CMOS camera (Nikon), and 37 °C, 5% $CO_2$ heat stage (Live Cell Instrument). Imaging was acquired using Nikon element AR software. Images with multiple z-stacks were processed with Huygens deconvolution (Scientific Volume Imaging), and the maximum intensity projections of images were produced by Nikon element AR software (Nikon). The image analysis was mainly conducted by Nikon element AR software (Nikon).

**Measurement of IFT dynamics**. Neon-IFT88 or Neon-Kif3B-stable NIH3T3 cells were transfected with the indicated constructs and then seeded onto poly(D-lysine)-coated borosilicate glass Lab-Tek eight-well chambers (Thermo Scientific). The cells were imaged every 200 ms for 30 s on a Nikon T1 inverted fluorescence microscope (Nikon) with a ×60 oil objective (Nikon), DS-Qi2 CMOS camera (Nikon), and 37 °C, 5% $CO_2$ heat stage (Live Cell Instrument). Time-lapse images were processed by Huygens deconvolution (Scientific Volume Imaging), and kymographs were produced with ImageJ and the plug-in KymographClear[70].

**Immunofluorescence staining**. Cells were plated on poly(D-lysine)-coated borosilicate glass Lab-Tek eight-well chambers (Thermo Scientific), fixed in 4% paraformaldehyde (Electron Microscopy Sciences) at room temperature for 10 min, and then in 100% methanol (Sigma-Aldrich) at −20 °C for 4 min. Fixed samples were permeabilized by 0.1% Triton X-100 (Sigma-Aldrich) and then incubated in blocking solution (PBS with 2% bovine serum albumin) for 30 min at room temperature. To label glutamylated tubulin in the cytosol and basal body, cells were cold-treated at 4 °C for 1 h and then fixed in cold methanol at −20 °C for 10 min. Primary antibodies were diluted in blocking solution and were used to stain the cells for 1 h at room temperature. The primary antibodies used in this study were anti-glutamylated tubulin (1:500 dilution; AG-20B-0020, AdipoGen), anti-Gli3 (1:200 dilution; AF3690, R&D Systems), anti-acetylated tubulin (1:500 dilution, T7451, Sigma-Aldrich), anti-Smoothened (1:100 dilution, ab 38686, Abcam), anti-Arl13B (1:500 dilution, ab83879, Abcam), anti-polyglutamate chain (1:500 dilution, IN105, AdipoGen), anti-Δ2-tubulin (1:500 dilution, AB3203, Merck Millipore), anti-detyrosinated tubulin (1:500 dilution, AB3201, Millipore), and anti-GPR161 (1:200 dilution; 13398-1AP, Proteintech). Secondary antibodies were diluted in blocking solution (1:1000 dilution) and were incubated with samples for 1 h at room temperature.

**FRAP experiments**. FRAP experiments were carried out with a Nikon A1 confocal system (Nikon) and ×100 oil objective (Nikon). Before bleaching, two sequential images were taken to obtain a baseline of fluorescence intensity of Neon-IFT88 or GFP–Gli3. The cilia region of cells expressing GFP–Gli3 or Neon-IFT88 was then photobleached and allowed to recover for 20 min. The fluorescence intensity of GFP–Gli3 or Neon-IFT88 was measured with Nikon element AR software (Nikon).

**Statistical analysis**. Statistical analysis was performed with an unpaired two-tailed Student's *t*-test, and whether variances were equal or not was determined by the F-test. *P* values were calculated, when $P \geq 0.05$ represents no significant difference, $P < 0.05$ represents a significant difference, and $P < 0.01$ represents a highly significant difference.

**Data availability**. The data that support the findings of this study are available from the corresponding author upon reasonable request.

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

## Acknowledgements

We thank Dr. W. James Nelson (Stanford University) for the MAP4m construct, Dr. Gregory J. Pazour (University of Massachusetts Medical School) for the IFT88 and Kif3B constructs, Dr. Maarten F. Bijlsma and Dr. Helene Damhofer (University of Amsterdam) for the GBS–GFP construct, Dr. Koji Ikegami (Hamamatsu University School of Medicine) for the CCP5 construct, Dr. Carsten Janke (Institut Curie) for the TTLL6 construct, and Dr. Jin-Wu Tsai (National Yang-Ming University) for the Patched1-YFP construct. We also thank Dr. Kristen Verhey (University of Michigan Medical School) for helpful discussions. We are grateful to Dr. Koji Ikegami (Hamamatsu University School of Medicine) for critical reading of the manuscript, as well as Robert DeRose for writing suggestions. We thank Emily Su (Johns Hopkins University) for assistance with experiments and Dr. Tasuku Ueno (University of Tokyo) for the synthesis of GA₃-AM. This study was supported in part by the National Institutes of Health (GM105448 and GM118082 to R.R.; GM109984 to N.C.S.; R01DK102910 to T.I.), the Ministry of Science and Technology (MOST), Taiwan (MOST 104-2311-B-007-001, MOST 105-2628-B-007-001-MY3, and Program for Translational Innovation of Biopharmaceutical Development–Technology Supporting Platform Axis–nMACS imaging No. 107-0210-01-19-04 to Y.C.L.), and start-up funding from National Tsing Hua University to Y.C.L.

## Author contributions

S.R.H., C.L.W., Y.S.H., Yu-Chen Chang, and Y.C.L. generated DNA constructs, and S.R.H., C.L.W., Y.S.H., Y.C.L., Yu-Chen Chang, Ya-Chu Chang, C.Y.L., N.H., H.C.C., Yueh-Chen Chiang, and W.E.H. performed cell biology experiments and quantified the imaging results. Y.C.L. and C.L.W. performed live-cell imaging. N.C.S. generated the Neon plasmid. G.V.P. established the NIH3T3:8xGBS–GFP reporter line under the supervision of R.R., S.R.H., C.L.W., and T.I., and Y.C.L. wrote the paper.
