## [Peer Review File · Nature Communications]

Reviewer #1 (Remarks to the Author):

Comments to the authors:

In this manuscript, the authors have developed a novel methodology to express and direct the deglutamylase CCP5 and its active site domain (CCP5CD) into primary cilia. The concept of using Rapamycin-based dimerization to achieve a precise localization of tubulin-modifying enzymes into cilia is unique and could have a strong impact on future work deciphering the role of a whole range of tubulin PTMs in cilia. Especially the fact that the authors observe almost a complete loss of glutamylation only 30 min after Rapamycin-mediated induction of the system indicates a fast kinetics of PTM modulation, which could previously not be obtained with simply overexpressing modifying enzymes. The beauty of the new system is that tubulin glutamylation can be changed at any point after ciliogenesis, and thus the role of this PTM in ciliogenesis and ciliary functions can be separated, which is what the authors did next.

Using their system, the authors study the role of glutamylation in primary cilia, and confirm the importance of this PTM in cilia biogenesis. Strikingly however they can now also show (using the inducible system) that glutamylation has no impact on the length of cilia once they are assembled. The authors next interrogated the role of polyglutamylation in already assembled cilia, and demonstrated an involvement in Hedgehog signaling: when active CCP5 is localized to cilia, they found an absence of the localization of Hedgehog proteins Smo and Gli as a result of reduced polyglutamylation, indicating a dysfunction of the intra-flagellar transport (IFT).

In summary, the current work is solid and gives new information into the role of tubulin PTMs in primary cilia. However, there are several concerns that need to be addressed before the manuscript can be considered for publication.

1. The text contains a number of overstatements and unprecise writing, and thus need rewriting. The most striking examples are listed below, however the authors should carefully re-read the entire manuscript to avoid unprecise statements.

Examples:

- abstract line 1; page 3 line 5: "A bundle of microtubules called an axoneme..." – a bundle suggests a disordered structure, which is the opposite of what an axoneme is.

- line 4: "This has led to the concept of a tubulin code," – overstatement: not the PTMs in the axoneme, but the general diversity of tubulin in all cells of eukaryotic mechanisms led to the formulation of the tubulin code hypothesis.

- line 7: "To crack the tubulin code..." – overstatement, the authors just want to alter tubulin glutamylation in cilia, and not crack the entire tubulin code.

- line 8: "...rewrite tubulin PTMs..." – overstatement: the authors will just remove tubulin glutamylation in the current work.

- line 9 "... de novo deglutamylation..." – deglutamylation is NOT a modification per se, so it cannot be generated de novo.

- page 3 line 7: "The C-terminal tails of axonemal tubulin are covalently linked with multiple glutamate side chains through polyglutamylation." – hard to understand: better state that polyglutamylation generates glutamate chains at the C-terminal tails of tubulin.

- page 4 line 1: it should be mentioned that some glutamylases also have other substrates than tubulin.

- page 4 line 5: it is true that Berezniuk et al. show that under in vitro conditions, CCP5 is able to shorten long glutamate chains, however this does not exclude that this activity is not predominant in vivo. This should be discussed in the light of the following publication, which shows that this is indeed the case: Wu H-Y, Wei P, Morgan JI (2017) Role of Cytosolic Carboxypeptidase 5 in Neuronal Survival and Spermatogenesis. *Sci Rep* 7: 41428

- page 4 line 12: The authors provide a list of how tubulin glutamylation was studied so far, and completely forgot the multiple publications on TTLL- and CCP-KO animals that have been published over the last years.

- page 4 line 16: spastin is not recruited to microtubules by polyglutamylation, but it is activated on the microtubules. Spastin alone can localize to microtubules via a specific microtubule binding domain. This has been shown in the ref 15 which the authors cite. They should additionally cite

Valenstein ML, Roll-Mecak A (2016) Graded Control of Microtubule Severing by Tubulin Glutamylation. *Cell* 164: 911-921.

- page 14 line 4: "Polyglutamylation on the surface of axonemes serves as an interface between microtubules and IFT motors." – overstatement: this has never been shown. What has been shown is that polyglutamylation somehow alters certain interactions between IFT motors and axonemal microtubules. The interface of many microtubule motors is NOT the C-terminal tail, where glutamylation takes place.

2. The authors have constructed a set of constructs derived from the deglutamylase CCP5, which they attract to the cilia by dimerizing them with the FRB/FKRB rapamycin system with the microtubule-associated protein MAP4. This system should allow to focus CCP5 into the cilia and thus lead to a specific deglutamylation only of the ciliary microtubules – the axonemes. How do the authors make sure that the active CCP5 constructs do not deglutamylate cellular substrates such as microtubules and other glutamylated proteins BEFORE they localize the enzyme to cilia? Obviously in standard cultured cell lines, there is little glutamylation on interphase microtubules, but there are ways of visualizing even weak glutamylation levels by using elevated concentrations of anti-glutamylation antibodies (see Magiera MM, Janke C (2013) Investigating tubulin posttranslational modifications with specific antibodies. In *Methods Cell Biol*, Correia JJ, Wilson L (eds), Vol. 115, 2013/08/27 edn, pp 247-267. Burlington: Academic Press).

3. In Fig. 1e, the authors show that 30 min after induction of ciliary deglutamylation with Rapamycin, the CCP5CD-Neon-FKBP recruited to the cilia is able to deglutamylate almost all of the glutamylated tubulin, except the proximal end of the cilia. The rapamycin, they still residual glutamylation at the base of the cilia. They suggest that this could be due to the activity of endogenous TTLL glutamylases present at the base of the cilia. However, considering that CCP5 is overexpressed as compared to the endogenous TTLLs, why do the authors think that CCP5 could not overcome this supposedly much lower endogenous activity? They should check the glutamylation status of the ciliary base at later time points after CCP5 induction to verify if this glutamylation remains.

4. In Fig 2, the authors aim to show that STRIP does not alter the average length of the cilium. However their quantification shows that the average length in the CCP5CD-Neon-FKBP cells is 2 μm , while it is 2.5 μm in the Neon-only cells. This represents a 20% change in ciliary length, and it might turn out significant if the authors would measure more cilia. To clarify this point, a more thorough analysis (measuring more cilia, proper statistic analyses) is required to elucidate the role of CCP5 activity, and thus glutamylation, in ciliary length control.

5. At page 13, the authors write that the steady-state of primary cilia structure is independent of the glutamylation state. As it has already been established that the other tubulin posttranslational modification, glycylation, is linked to glutamylation and plays a role in stabilizing cilia (Bosch Grau, M et al. 2013. Tubulin glycylation and glutamylation have distinct functions in stabilization and motility of ependymal cilia. *J Cell Bio* 202: 441–45; Bosch Grau M et al. (2017) Alterations in the balance of tubulin glycylation and glutamylation in photoreceptors leads to retinal degeneration. *J Cell Sci* 130: 938-949), it will be necessary to check if glycylation is altered (increased?) in the presence of CCP5CD-Neon-FKBP.

6. In Fig. 3 the authors show that rapid tubulin deglutamylation resulting in loss of signal for Neon-IFT88 on the axonemes. However, when they measure the anterograde movement of Neon-IFT88, the reduction of IFT88 mobility is not very drastic: they determine a 0.15 sec/ μm slower transport in the absence of glutamylation. This stands in strong contrast to their immunofluorescence images, in which hardly any IFT88 is observed in cilia, and where IFT88 accumulates at the basal bodies. These two results are somewhat contradictory, and raise the question if the authors have looked into the IFT88 movement after a longer period of time after Rapamycin induction. Moreover, the authors should check if IFT88 does not enter the cilium at all, and if this is the case, if there is a reduced rate of ciliary entry that could explain the quasi-absence of IFT88 from deglutamylation cilia.

7. In Fig. 3g, the authors show a redistribution of Kif3B between the base and the rest of the cilia after deglutamylation. They show an enrichment of Kif3B from about 60% in control to about 70% in deglutamylation cilia, and state: “In summary, these results support that glutamylation controls anterograde IFT through Kif3B of the kinesin-2 complex.” (intersection of pages 15 & 16). Considering the small change in localization, they might want to discuss why they think there is only such a subtle effect, considering their strong claim that the mechanism of control is a regulation of the Kif3b motor.

They in fact discuss this in the discussion (intersection of pages 18 and 19: “Another explanation is that residual anterograde IFT in the STRIP cells may have been sufficient for cilia growth and maintenance, unlike the cases for near complete loss-of-function of kinesin-2 proteins”. Thus there is no need of such a strong conclusion in the result section

8. In Fig. 4 the authors show that the localization of the signaling components Smo3 and Gli is impaired when axonemes are deglutamylation by the CCP5CD-Neon-FKBP upon Rapamycin induction. They also show that Gli is localized towards the tip of the cilium, whereas Smo localizes throughout the cilium. How do the authors explain the loss of the Gli signal from the tip of the cilium upon deglutamylation?

Reviewer #2 (Remarks to the Author):

The manuscript by Wang et al describes a new approach to altering tubulin modifications in cells in a controlled manner. The authors utilize rapamycin-induced recruitment of a deglutamyase, CCP5, to the axoneme and show that glutamylation is reduced. Furthermore, anterograde IFT and Hedgehog signaling are impaired. This is an interesting approach that will be useful in the field. I have some concerns that need to be addressed before publication.

The approach is interesting and will be very useful if it is generally applicable. Can other CCPs or TTLs be recruited to the axoneme to alter tubulin modifications?

A major advantage of the assay is the ability to alter tubulin modifications in a rapid and spatially-specific manner. But since the truncated CCP5CD is itself an active enzyme, it can presumably alter glutamylation patterns when expressed in cells. Does it alter the glutamylation of cytoplasmic microtubules? The truncated CCP5CD presumably must cycle through the cilium in order to be trapped by MAP4. Thus, why doesn't it reduce glutamylation of axonemes when expressed, even without the rapamycin treatment? Higher magnification images of the effects of CCP5CD before and after rapamycin treatment need to be shown. Acetylated tubulin is likely not a good marker for other effects of the CCP5 since acetylation is inside the microtubule. What are the effects on detyrosination?

Rapamycin-induced anchoring of YFP-FKBP to the axoneme occurred within 2 min (Fig 1c,d) whereas anchoring of CCP5CD-Neon-FKBP takes much longer (20-30 min?, Fig 2d). Please provide a time course of the CCP5CD recruitment to the axoneme. Why is the recruitment of CCP5CD so much slower?

Does the residual glutamylation at the base of the cilium localize to the centriole and/or at the transition zone?

The data in Figure 2a-e suggest that glutamylation plays no role in cilia maintenance. However, the suggestion that loss of glutamylation alters anterograde IFT (Figure 3) appears to be at odds with this finding. It seems likely that the time course of the experiment in Figure 2a-e is not sufficient for the authors to conclude anything about the role of glutamylation in cilia maintenance. In particular, work in other systems, particularly *Chlamydomonas*, has shown that it takes many hours of disrupted IFT for the cilia to go away (e.g. Kozminski et al. 1995 JCB, Engel et al. 2012 JCB; Lin et al.

2013 Cilia). Thus, the experiments in Figure 2 need to be repeated with a longer time course of STRIP. In addition, the fact that full length CCP5 localizes to and alters the glutamylation of cilia suggest an easier experiment where expression of CCP5 should prevent ciliation or cause resorption of cilia that are already formed.

The data on IFT response to changes in glutamylation are not convincing. I have a hard time seeing any anterograde IFT in Figure 2C so I am not convinced about these data. The anterograde lines that are drawn for the kymograph in Figure 2c suggest that the speed of anterograde IFT is dramatically reduced upon rapamycin treatment but the change displayed in Figure 3d middle does not reflect this. Are the images in Figure 2c representative?

The authors demonstrate that reduced glutamylation upon rapamycin treatment alters anterograde IFT (Figure 3), localization of Hedgehog components to the cilium (Figure 4), and transcriptional response to Hedgehog (Figure 4). Based on this, the authors propose that glutamylation alters the motility of IFT motors. An alternative explanation is that rapamycin induced trapping of CCP5CD at MAP4-rich microtubules alters the structure or function of the transition zone and thus the entrance of proteins into the cilium. This could explain the effects on IFT in Figure 3, and trafficking of Hedgehog components in Figure 4. Does the number of anterograde and retrograde IFT events change after rapamycin treatment? It looks like the IFT trains are stuck at the base of the cilium and unable to enter after rapamycin treatment. Does the turnover of Hedgehog pathway components in the cilium change after rapamycin, for example using FRAP analysis of Gli3 at the cilium tip? Does the exit of Hedgehog components Patched or GPR161 change upon rapamycin treatment? If decreased glutamylation is specifically affecting anterograde IFT, then there should be no effect on exit of Patched or GPR161.

For all graphs, what is being displayed in the y-axis needs to be clarified. For example, for Figure 1f, the figure legend states that fluorescence intensities were measured but the y-axis of the graph indicates average length. Average length of what? Quantification of the changes in both fluorescence intensities and extent of the modifications is critical for the reader to assess the effects of CCP5 recruitment. Why is a normalized cilia length used in Figure 2e rather than the average length as in all the other figures? For all bar graphs, the spread of data across the population needs to be shown using box-and-whisker or dot plots.

Reviewer #3 (Remarks to the Author):

The manuscript by Lin et al aims at studying the role of glutamylation in ciliogenesis, ciliary maintenance, and signaling. The authors devise a clever approach of forced dimerization of a deglutamylase with an axonemal-targeted MAP. The approach is well characterized using two dimerizers (although the Gibberlin-based system is not used in their experiments later), and the authors find that a limited catalytic domain is captured in the cilia by a FRB-MAP fusion that predominantly localizes to axonemes. The authors have carefully established the validity of their experimental system by demonstrating that (a) other tubulin modifications in axonemes are not impacted, and (b) glutamylation in other cellular locations including cytosol are impacted by the full length CCP5, but not by the isolated catalytic domain. Although the authors have nicely established a novel experimental system for studying glutamylation, which by itself is an important method for the field, I am not convinced by the three conclusions that the authors propose in the current form of the manuscript. I explain reasons for my lack of conviction below:

1. Forced CCL5CD localization in cilia prevents ciliogenesis. I am struck by the consistency in cilia length in unstarved unsynchronized cells (<1 μm ; Fig 2G-H). I am not sure how the authors did the experiment in this case, as 3T3 cells ciliate even in the presence of serum, and/or if they selected for cilia of a certain length.

The authors next measure increase in ciliary length upon starvation for 4h, and notice less increase in case of CCP5CD expression. It is important to emphasize here whether Rapamycin is reversible in its action, so that the dimerization is lost upon changing from Rapamycin containing medium into starvation medium, and that the effects the authors see are because of forced localization of the CCL5CD before initiating starvation. In experiments like these (also in Fig 4), it is important to show how long it takes for CCP5CD to leave the cilia after removing Rapamycin.

2. Anterograde IFT rates are reduced upon forced CCP5CD localization. I am unable to assess the data provided, given the current quality of movies in supplement, and the gross discrepancy of reported anterograde and retrograde velocities with respect to published literature (for e.g. see PMC4610257, where the anterograde and retrograde IFT velocities in kidney collecting duct cells are close to 1 $\mu\text{m}/\text{s}$). The reported anterograde and retrograde IFT velocities here are $\sim 0.5 \mu\text{m}/\text{s}$ and $\sim 0.3 \mu\text{m}/\text{s}$, respectively, which are very low. The designated lines in kymograph (for e. g., horizontal anterograde lines and thick retrograde lines in Fig 3C) are most likely not moving particles. I strongly recommend IMCD cells for these experiments, as the cilia are longer than 3T3 cells, and there is available good quality data to compare.

3. Reduced Shh signaling upon forced CCP5CD localization. My concerns with this segment of the paper comes from the last panel, where the authors show that increased Gli1 levels in nucleus are reduced upon forced CCP5CD localization in SAG-treated cells. The cilia in these GBS-GFP-3T3 cells are quite long (ranging up to 20 μm , compared to 5 μm in parental cells), and thus given the low increase in Gli1 levels in nucleus in control cells, I am unable to interpret the data. The authors

should again discuss the role of reversibility of the dimerization upon SAG treatment, and how long the CCP5CD resides in cilia after Rapamycin is removed.

Finally, an orthogonal approach, such as RNAi for a glutamylase to validate some of the conclusions would strengthen the data.

Minor comments:

1. Please describe methods in legends to sufficiently understand the figures. For e.g. in Fig 2 and 4, were the cells confluent before starving?

2. Cilia lengths vary quite a bit between figures although the authors use 3T3 cells. For e. g cilia lengths are ~10 μM (Fig 1C, 2D), ~5 μM (Fig 1E, 2B), ~20 μM (Fig. 4f). Please explain why, and if the different lengths impact on interpretation of data. Please explain the term “normalized cilia length” in Fig 2E.

3. Velocities in Fig 3 to be denoted as “ $\mu\text{m/s}$ ” not “ $\text{s}/\mu\text{m}$ ”

4. Writing needs to be improved throughout the text. Some examples include (a) rewriting parts of the abstract, (b) Page 11 (top line, add “alteration in” cilia length), Page 16 (change “The Lechtreck group once showed”), etc.

5. Methods must include plasmids and/or if P2A-based co-expressing construct was used in all experiments. This is important for data interpretations, as equivalent amounts of FKB-FRB fusions are needed for the experiments.

6. Legends should include transfection constructs, cell numbers or confluence before starvation etc for the sake of reproducing the data by other labs.

7. Methods should mention % of transfected cells in 3T3 cells, and how transfected cells were selected for data collection (in many cases data collection being from <10 total cells from 3-4 experiments).

8. Probably, normalization of the fluorescent intensities of CCP5CD-FKBP signal with respect to the MAP-FRB signal is necessary to interpret the data in most figures.

We thank the reviewers and the editors for the positive notes and critical suggestions. We have revised the manuscript accordingly and have provided a separate manuscript file with tracked changes. The reviewers' comments are itemized below, followed by our point-by-point responses.

Major changes we have made, including new items:

- 1) Fig. 1d: According to reviewers' suggestion, the intensity measured for YFP-FKBP was divided by that for CFP and plotted.
- 2) Fig. 2c: We have repeated the experiments for measuring cilia length in the three different groups. The results are consistent with our previous finding that deglutamylation does not affect cilia length.
- 3) Fig. 4a: This now shows the maximal intensity projections of Neon-IFT88 before and after rapamycin treatment.
- 4) Fig. 4b: Linescan profiles of Neon-IFT88 in Fig. 4a are now shown.
- 5) Fig. 4c: To improve the quality of the kymograph, time-lapse imaging of Neon-IFT88 in Fig. 4a was utilized to generate kymograph using the software "KymographClear" (see Methods). Red, green, and blue lines represent the trajectories of Neon-IFT88 particles in the anterograde and retrograde directions and static Neon-IFT88, respectively.
- 6) Fig. 4d: Box plots were used to show the velocity of IFT88 under the indicated conditions.
- 7) Fig. 5: We added a FRAP analysis of Neon-IFT88 in control and deglutamylated cilia.
- 8) Fig. 7f: To clearly indicate that the GFP signal represents Gli activity, we have added "(Gli activity)" to the figure.
- 9) Supplementary Fig. 1c: Box plots were used to show the average length of cilia labeled by the indicated antibodies.
- 10) Supplementary Fig. 1d: To improve the quality of the kymograph, the time-lapse imaging of Neon-IFT88 in Supplementary Fig. 1d was utilized to generate kymograph using KymographClear (see Methods). Red, green, and blue lines represent the trajectories of Neon-IFT88 particles in the anterograde and retrograde directions and static Neon-IFT88, respectively.
- 11) Supplementary Fig. 1e: Box plots were used to show the velocity of IFT88 under the indicated conditions.
- 12) Supplementary Fig. 2c: The YFP-FKBP fluorescence intensity was divided by that of CFP and plotted.
- 13) Supplementary Fig. 3c: Box plots were used to show the average length of cilia labeled by the indicated antibodies.
- 14) Supplementary Fig. 4c: Box plots were used to show the average length of cilia labeled by the indicated antibodies.
- 15) Supplementary Fig. 5: The rapamycin-induced translocation of CCP5CD-Neon-FKBP was added.
- 16) Supplementary Fig. 6: We added the distribution of residual glutamylated tubulin after deglutamylation treatment.
- 17) Supplementary Fig. 7b: Box plots were used to show the average length of cilia labeled by the indicated antibodies.
- 18) Supplementary Fig. 7e: The effect of deglutamylation on tubulin deetyrosination was added.
- 19) Supplementary Fig. 7d: Box plots were used to show the average length of cilia labeled by the indicated antibodies.
- 20) Supplementary Fig. 7e: The effect of deglutamylation on the extent of tubulin deetyrosination was added.
- 21) Supplementary Fig. 7f: Box plots were used to show the average length of cilia labeled by the indicated antibodies.
- 22) Supplementary Fig. 8b: Box plots were used to show the average length of glutamylated axoneme and average fluorescence intensity of glutamylated tubulin in the cytosol and basal body.

- 23) Supplementary Fig. 9b: Box plots were used to show the average length of cilia under different conditions.
- 24) Supplementary Fig. 11: We evaluated the effect of long-term deglutamylation on IFT dynamics.
- 25) Supplementary Fig. 12: To improve the quality of the kymograph, time-lapse imaging of Neon-Kif3B in Supplementary Fig. 12a was utilized to generate kymograph using KymographClear (see Methods). Red, green, and blue lines represent the trajectories of Neon-Kif3B particles in the anterograde and retrograde directions and static Neon-Kif3B, respectively.
- 26) Supplementary Fig. 13a,b: We evaluated the effect of deglutamylation on the distribution of patched1 protein.
- 27) Supplementary Fig. 13c,d: We evaluated the effect of deglutamylation on the distribution of GRP161 protein.
- 28) Supplementary Fig. 14: A FRAP analysis of Gli3 in deglutamyolated cilia was added.
- 29) Supplementary Movie 3: The translocation of CCP5CD-Neon-FKBP onto the axoneme upon rapamycin treatment was added.
- 30) Supplementary Movie 7: A FRAP analysis of Neon-IFT88 in control and deglutamyolated cilia was added.
- 31) Supplementary Movie 9: A FRAP analysis of GFP-Gli3 in control and deglutamyolated cilia was added.

Reviewers' comments:

Reviewer #1 (Remarks to the Author):

Comments to the authors:

In this manuscript, the authors have developed a novel methodology to express and direct the deglutamylose CCP5 and its active site domain (CCP5CD) into primary cilia. The concept of using Rapamycin-based dimerization to achieve a precise localization of tubulin-modifying enzymes into cilia is unique and could have a strong impact on future work deciphering the role of a whole range of tubulin PTMs in cilia. Especially the fact that the authors observe almost a complete loss of glutamylation only 30 min after Rapamycin-mediated induction of the system indicates a fast kinetics of PTM modulation, which could previously not be obtained with simply overexpressing modifying enzymes. The beauty of the new system is that tubulin glutamylation can be changed at any point after ciliogenesis, and thus the role of this PTM in ciliogenesis and ciliary functions can be separated, which is what the authors did next.

Using their system, the authors study the role of glutamylation in primary cilia, and confirm the importance of this PTM in cilia biogenesis. Strikingly however they can now also show (using the inducible system) that glutamylation has no impact on the length of cilia once they are assembled. The authors next interrogated the role of polyglutamylation in already assembled cilia, and demonstrated an involvement in Hedgehog signaling: when active CCP5 is localized to cilia, they found an absence of the localization of Hedgehog proteins Smo and Gli as a result of reduced polyglutamylation, indicating a dysfunction of the intra-flagellar transport (IFT).

In summary, the current work is solid and gives new information into the role of tubulin PTMs in primary cilia. However, there are several concerns that need to be addressed before the manuscript can be considered for publication.

We cannot appreciate enough to this reviewer for not only piercing the values of our approach, but also offering constructive expert comments which could greatly help us improve the manuscript. We made our best effort to follow the suggestions.

1. The text contains a number of overstatements and unprecise writing, and thus need rewriting.

The most striking examples are listed below, however the authors should carefully re-read the entire manuscript to avoid unprecise statements.

Examples:

- abstract line 1; page 3 line 5: “A bundle of microtubules called an axoneme...” – a bundle suggests a disordered structure, which is the opposite of what an axoneme is.

We have changed the original sentence “A bundle of microtubules called an axoneme serves as the...” to “The ciliary axoneme serve as the....”. (Page 2, Line 2).

- line 4: “This has led to the concept of a tubulin code,” – overstatement: not the PTMs in the axoneme, but the general diversity of tubulin in all cells of eukaryotic mechanisms led to the formulation of the tubulin code hypothesis.

We have removed the sentence “This has led to the concept of a tubulin code where each unique combination of PMTs...”.

- line 7: “To crack the tubulin code...” – overstatement, the authors just want to alter tubulin glutamylation in cilia, and not crack the entire tubulin code.

We have changed the original sentence “To crack the tubulin code...” to “To address a role of glutamylation in cilia,...” (Page 2, Line 8).

- line 8: “...rewrite tubulin PTMs...” – overstatement: the authors will just remove tubulin glutamylation in the current work.

We have changed the original sentence “...rewrite tubulin PTMs...” to “...deplete tubulin glutamylation...”. (Page 2, Line 8).

- line 9 “... de novo deglutamylation...” – deglutamylation is NOT a modification per se, so it cannot be generated de novo.

We have changed the original sentence “...de novo deglutamylation...” to “...rapid deglutamylation...”. (Page 2, Line 10).

- page 3 line 7: “The C-terminal tails of axonemal tubulin are covalently linked with multiple glutamate side chains through polyglutamylation.” – hard to understand: better state that polyglutamylation generates glutamate chains at the C-terminal tails of tubulin.

We appreciate the suggestion and have changed the original sentence to “polyglutamylation generates glutamate chains at the C-terminal tails of tubulin”. (Page 3, Line 8).

- page 4 line 1: it should be mentioned that some glutamylases also have other substrates than tubulin.

We now describe this as:

...beside tubulin, many nucleocytoplasmic shuttling proteins such as nucleosome assembly protein are also identified as the substrates of glutamylases^{10,35}. Therefore, global manipulation of genes encoding enzymes that modulate glutamylation is insufficient to specifically perturb axonemal glutamylation. (Page 5, Line 14)

Relevant references:

10. van Dijk, J. et al. A Targeted Multienzyme Mechanism for Selective Microtubule Polyglutamylation. *Mol. Cell* 26, 437–448 (2007).
35. Regnard, C. Polyglutamylation of Nucleosome Assembly Proteins. *J. Biol. Chem.* 275, 15969–15976 (2000).

- page 4 line 5: it is true that Berezniuk et al. show that under in vitro conditions, CCP5 is able to shorten long glutamate chains, however this does not exclude that this activity is not predominant in vivo. This should be discussed in the light of the following publication, which shows that this is indeed the case: Wu H-Y, Wei P, Morgan JI (2017) Role of Cytosolic Carboxypeptidase 5 in Neuronal Survival and Spermatogenesis. *Sci Rep* 7: 41428

We have added the reference “Wu *et al.* (2017)” at the end of the sentence “CCP5 preferentially removes a glutamate at the branching fork, whereas other CCP members target a glutamate residue in a linear, tandem sequence in vivo”. (Page 4, Line 5)

Relevant reference:

13. Wu, H.-Y., Wei, P. & Morgan, J. I. Role of Cytosolic Carboxypeptidase 5 in Neuronal Survival and Spermatogenesis. *Sci. Rep.* 7, 41428 (2017)

- page 4 line 12: The authors provide a list of how tubulin glutamylation was studied so far, and completely forgot the multiple publications on TLL- and CCP-KO animals that have been published over the last years.

We have added the relevant references as below.

Relevant references:

19. Kubo, T., aki Yanagisawa, H., Yagi, T., Hirono, M. & Kamiya, R. Tubulin Polyglutamylation Regulates Axonemal Motility by Modulating Activities of Inner-Arm Dyneins. *Curr. Biol.* 20, 441–445 (2010).
20. Pathak, N., Austin, C. A. & Drummond, I. A. Tubulin tyrosine ligase-like genes *tll3* and *tll6* maintain zebrafish cilia structure and motility. *J. Biol. Chem.* 286, 11685–11695 (2011).
21. Chen, D. et al. The zebrafish *flee* gene encodes an essential regulator of cilia tubulin polyglutamylation. *Mol. Biol. Cell* 19, 308–317 (2007).
22. Lyons, P. J., Sapio, M. R. & Fricker, L. D. Zebrafish cytosolic carboxypeptidases 1 and 5 are essential for embryonic development. *J. Biol. Chem.* 288, 30454–30462 (2013).
23. Suryavanshi, S. et al. Tubulin Glutamylation Regulates Ciliary Motility by Altering Inner Dynein Arm Activity. *Curr. Biol.* 20, 435–440 (2010).
24. Ikegami, K., Sato, S., Nakamura, K., Ostrowski, L. E. & Setou, M. Tubulin polyglutamylation is essential for airway ciliary function through the regulation of beating asymmetry. *Proc. Natl. Acad. Sci. U. S. A.* 107, 10490–10495 (2010).
25. Lee, J. E. et al. CEP41 is mutated in Joubert syndrome and is required for tubulin glutamylation at the cilium. *Nat. Genet.* 44, 193–199 (2012).
26. Alford, L. M. et al. The nexin link and B-tubule glutamylation maintain the alignment of outer doublets in the ciliary axoneme. *Cytoskeleton* 73, 331–340 (2016).
27. Grau, M. B. et al. Tubulin glycylation and glutamylation have distinct functions in stabilization and motility of ependymal cilia. *J. Cell Biol.* 202, 441–451 (2013).
28. Kim, J. et al. Functional genomic screen for modulators of ciliogenesis and cilium length. *Nature* 464, 1048–1051 (2010).
29. Konno A, Ikegami K, Konishi Y, Yang HJ, Abe M, Yamazaki M, Sakimura K, Yao I, Shiba K, I. K. and S. M. Doublet 7 shortening, doublet 5-preferential poly-Glu reduction, and beating stall of sperm flagella in *Tll9^{-/-}* mice. *J. Cell Sci.* pii:jcs.185983 (2016).
30. Kubo, T., Yagi, T. & Kamiya, R. Tubulin polyglutamylation regulates flagellar motility by controlling a specific inner-arm dynein that interacts with the dynein regulatory complex. *Cytoskeleton* 69, 1059–1068 (2012).
31. Lee, G. S. et al. Disruption of *Tll5/Stamp* gene (tubulin tyrosine ligase-like protein 5/SRC-1 and TIF2-associated modulatory protein gene) in male mice causes sperm malformation and infertility. *J. Biol. Chem.* 288, 15167–15180 (2013).
32. O’Hagan, R. et al. The tubulin deglutamylase CCPP-1 regulates the function and stability of sensory cilia in *C. elegans*. *Curr. Biol.* 21, 1685–1694 (2011).

33. Vogel, P., Hansen, G., Fontenot, G. & Read, R. Tubulin tyrosine ligase-like 1 deficiency results in chronic rhinosinusitis and abnormal development of spermatid flagella in mice. *Vet. Pathol.* 47, 703–12 (2010).

- page 4 line 16: spastin is not recruited to microtubules by polyglutamylation, but it is activated on the microtubules. Spastin alone can localize to microtubules via a specific microtubule binding domain. This has been shown in the ref 15 which the authors cite. They should additionally cite Valenstein ML, Roll-Mecak A (2016) Graded Control of Microtubule Severing by Tubulin Glutamylation. *Cell* 164: 911-921.

We have added the reference and corrected the original sentence “the overexpression of the polyglutamylase TTL6 in HeLa cells leads to microtubule disassembly due to the recruitment of a severing enzyme, namely spastin,...” to “Tubulin hyperglutamylation leads to microtubule disassembly owing to the binding of a severing enzyme...”. (Page 4, Line 14)

Reference:

17. Valenstein, M. L., Roll-mecak, A., Valenstein, M. L. & Roll-mecak, A. Graded Control of Microtubule Severing by Tubulin Glutamylation. *Cell* 164, 911–921 (2016).

- page 14 line 4: “Polyglutamylation on the surface of axonemes serves as an interface between microtubules and IFT motors.” – overstatement: this has never been shown. What has been shown is that polyglutamylation somehow alters certain interactions between IFT motors and axonemal microtubules. The interface of many microtubule motors is NOT the C-terminal tail, where glutamylation takes place.

We have changed the original sentence “Polyglutamylation on the surface of axoneme serves as an interface between microtubules and IFT motors, suggesting the possibility that...” to “Besides mechanical support, axoneme also serve as railways for IFT. We thus evaluated whether axonemal deglutamylation impacts the rate of IFT”. (Page 14, Line 15)

2. The authors have constructed a set of constructs derived from the deglutamylase CCP5, which they attract to the cilia by dimerizing them with the FRB/FKRB rapamycin system with the microtubule-associated protein MAP4. This system should allow to focus CCP5 into the cilia and thus lead to a specific deglutamylation only of the ciliary microtubules – the axonemes. How do the authors make sure that the active CCP5 constructs do not deglutamylate cellular substrates such as microtubules and other glutamylated proteins BEFORE they localize the enzyme to cilia? Obviously in standard cultured cell lines, there is little glutamylation on interphase microtubules, but there are ways of visualizing even weak glutamylation levels by using elevated concentrations of anti-glutamylation antibodies (see Magiera MM, Janke C (2013) Investigating tubulin posttranslational modifications with specific antibodies. In *Methods Cell Biol*, Correia JJ, Wilson L (eds), Vol. 115, 2013/08/27 edn, pp 247-267. Burlington: Academic Press).

Cytosolic glutamylated tubulin can be clearly detected after cold treatment (Supplementary Fig. 8). Expression of CCP5CD did not lead to a noticeable change in tubulin glutamylation compared with the control condition (Neon alone; Supplementary Fig. 8). Recruitment of CCP5CD to the axoneme specifically removed the glutamylation on the axoneme without affecting the level of tubulin glutamylation at other cellular sites including the basal body and cytosol (Supplementary Fig. 8). It is somewhat interesting that expression of soluble CCP5CD in cells had only marginal effect on glutamylation in the cytosol, despite a nature of free diffusion. We speculate that the expression level of cytosolic CCP5CD may be low enough to take an effect, and/or that endogenous polyglutamylases in the cytosol may counteract CCP5CD-mediated deglutamylation in the cytosol.

3. In Fig. 1e, the authors show that 30 min after induction of ciliary deglutamylation with Rapamycin, the CCP5CD-Neon-FKBP recruited to the cilia is able to deglutamylate almost all of the glutamylated tubulin, except the proximal end of the cilia. The rapamycin, they still residual

glutamylation at the base of the cilia. They suggest that this could be due to the activity of endogenous TLL glutamylases present at the base of the cilia. However, considering that CCP5 is overexpressed as compared to the endogenous TLLs, why do the authors think that CCP5 could not overcome this supposedly much lower endogenous activity? They should check the glutamylation status of the ciliary base at later time points after CCP5 induction to verify if this glutamylation remains.

To induce long-term deglutamylation in cilia, we forcefully anchored the CCP5 catalytic domain (CCP5CD) to the axoneme via fusion with MAP4m. Accumulation of CCP5CD-MAP4m in cilia constitutively eliminated axonemal glutamylation, which still could not remove the residual glutamylation at the proximal end of the cilia (Supplementary Fig. 4b). The residual glutamylation localized to the inversin zone but not the transition zone (Supplementary Fig. 6). The detailed mechanism of how axonemal tubulins in the inversin zone resist deglutamylation treatment is unclear.

4. In Fig. 2, the authors aim to show that STRIP does not alter the average length of the cilium. However their quantification shows that the average length in the CCP5CD-Neon-FKBP cells is 2 μm , while it is 2.5 μm in the Neon-only cells. This represents a 20% change in ciliary length, and it might turn out significant if the authors would measure more cilia. To clarify this point, a more thorough analysis (measuring more cilia, proper statistic analyses) is required to elucidate the role of CCP5 activity, and thus glutamylation, in ciliary length control.

We conducted two more independent experiments for cilia length measurement. The results show that the cilia length of cells expressing CCP5CD-Neon-FKBP was comparable with that of control cells after STRIP treatment ($2.33 \pm 0.19 \mu\text{m}$ in Neon-FKBP group vs. $2.39 \pm 0.16 \mu\text{m}$ in CCP5CD-Neon-FKBP group; $P = 0.81$; Fig. 2c).

5. At page 13, the authors write that the steady-state of primary cilia structure is independent of the glutamylation state. As it has already been established that the other tubulin posttranslational modification, glycylation, is linked to glutamylation and plays a role in stabilizing cilia (Bosch Grau, M et al. 2013. Tubulin glycylation and glutamylases have distinct functions in stabilization and motility of ependymal cilia. *J Cell Bio* 202: 441–45; Bosch Grau M et al. (2017) Alterations in the balance of tubulin glycylation and glutamylation in photoreceptors leads to retinal degeneration. *J Cell Sci* 130: 938-949), it will be necessary to check if glycylation is altered (increased?) in the presence of CCP5CD-Neon-FKBP.

We have used two commercially available antibodies against tubulin glycylation, namely anti-polyglycylated tubulin (AXO49, MABS276, Millipore) and anti-monoglycylated tubulin (TAP952, MABS277, Sigma-Aldrich), together with six different fixation protocols to label tubulin glycylation in primary cilia in NIH3T3 cells. Unfortunately, none of them could detect tubulin glycylation in primary cilia. Tubulin glycylation has been exclusively detected in motile cilia and flagella by using anti-TAP952 (Redeker *et al.*, *Science*, 1994; Rudiger *et al.*, *FEBS Lett*, 1995; Bre *et al.*, *Mol Biol Cell*, 1996; Weber *et al.*, *FEBS Lett*, 1996; Xia *et al.*, *J Cell Biol*, 2000). In the case of primary cilia, however, Dr. Janke and colleagues confirmed that anti-TAP952 failed to detect cilia glycylation in fibroblasts and other cell types (Rocha *et al.*, *EMBO J*, 2014; Gadadhar *et al.*, *J Cell Biol*, 2017). Very recently, the same group produced new glycylation-specific antibodies that enabled a clear detection of glycylation in very long primary cilia (>10 μm) but not (or rarely) in short primary cilia (<9 μm) (Gadadhar *et al.*, *J. Cell Biol*, 2017). Because the average length of primary cilia in NIH3T3 cells is usually <5 μm , it was challenging to detect tubulin glycylation in our system.

6. In Fig. 3 the authors show that rapid tubulin deglutamylation resulting in loss of signal for Neon-IFT88 on the axonemes. However, when they measure the anterograde movement of Neon-IFT88, the reduction of IFT88 mobility is not very drastic: they determine a 0.15 sec/ μm slower transport in the absence of glutamylation. This stands in strong contrast to their immunofluorescence images, in which hardly any IFT88 is observed in cilia, and where IFT88 accumulates at the basal bodies. These two results are somewhat contradictory, and raise the

question if the authors have looked into the IFT88 movement after a longer period of time after Rapamycin induction. Moreover, the authors should check if IFT88 does not enter the cilium at all, and if this is the case, if there is a reduced rate of ciliary entry that could explain the quasi-absence of IFT88 from deglutamylated cilia.

We realize that a single z-stack cannot fully represent the signal of Neon-IFT88. Therefore, in the revised manuscript, we show the imaging results by maximal intensity projection (Fig. 4a). Linescan analysis confirmed that Neon-IFT88 accumulates at the cilia base after STRIP treatment (Fig. 4b).

To study the effect of long-term deglutamylation on IFT, we measured the motility of Neon-IFT88 in cilia expressing CCP5CD-mCh-MAP4m (Supplementary Fig. 11). The results revealed that deglutamylation triggered by CCP5CD-mCh-MAPm slows anterograde IFT with no obviously effect on retrograde IFT, a result that is similar to the case for STRIP-induced rapid deglutamylation (Supplementary Fig. 11). Therefore, long-term deglutamylation does not have a more drastic effect on anterograde IFT compared with short-term treatment.

We used FRAP to check the ciliary entry and dynamics of IFT88 in normal and deglutamylated cilia. The results demonstrated that long-term deglutamylation slows the recovery of Neon-IFT88 in the axoneme without affecting the tethering of IFT88 to the basal body (Fig. 5). This suggests that deglutamylation decreases the motility of Neon-IFT88 along the axoneme in the anterograde direction. Together with kymographic results acquired for the slow anterograde IFT in deglutamylated cilia (Fig. 4d and Supplementary Fig. 11), we demonstrate that axonemal deglutamylation slows down anterograde IFT.

7. In Fig. 3g, the authors show a redistribution of Kif3B between the base and the rest of the cilia after deglutamylation. They show an enrichment of Kif3B from about 60% in control to about 70% in deglutamylated cilia, and state: "In summary, these results support that glutamylation controls anterograde IFT through Kif3B of the kinesin-2 complex." (intersection of pages 15 & 16). Considering the small change in localization, they might want to discuss why they think there is only such a subtle effect, considering their strong claim that the mechanism of control is a regulation of the Kif3b motor.

They in fact discuss this in the discussion (intersection of pages 18 and 19: "Another explanation is that residual anterograde IFT in the STRIP cells may have been sufficient for cilia growth and maintenance, unlike the cases for near complete loss-of-function of kinesin-2 proteins". Thus there is no need of such a strong conclusion in the result section

We have changed the original sentence "In summary, these results support that glutamylation controls anterograde IFT through Kif3B of the kinesin-2 complex" to "In summary, these results suggest that glutamylation controls anterograde IFT through Kif3B of the kinesin2-complex". (Page17 Line12)

8. In Fig. 4 the authors show that the localization of the signaling components Smo3 and Gli is impaired when axonemes are deglutamylated by the CCP5CD-Neon-FKBP upon Rapamycin induction. They also show that Gli is localized towards the tip of the cilium, whereas Smo localizes throughout the cilium. How do the authors explain the loss of the Gli signal from the tip of the cilium upon deglutamylation?

The kinesin-2 complex directly interacts with Hedgehog components such as Gli2 and Gli3 (Carpenter *et al.*, J Cell Sci, 2015). Moreover, knockout of IFT25, a component of the IFT-B complex, inhibits the trafficking of Gli to the cilia tip from the cytosol upon treatment of cells with Hedgehog ligand (Keady *et al.*, Dev Cell, 2012). These studies suggest that the anterograde IFT complex binds Hedgehog components and transports them to cilia where they regulate Hedgehog signaling. We therefore hypothesized that a deglutamylation-induced defect in anterograde IFT would affect the redistribution of Gli from the cytosol to the ciliary tip upon Hedgehog activation. Indeed, Our FRAP results confirmed that ciliary entry of Gli upon Hedgehog activation was impaired by deglutamylation (Supplementary Fig. 14), likely through an anterograde IFT-dependent mechanism (Fig. 7d).

Reviewer #2 (Remarks to the Author):

The manuscript by Wang et al describes a new approach to altering tubulin modifications in cells in a controlled manner. The authors utilize rapamycin-induced recruitment of a deglutamylase, CCP5, to the axoneme and show that glutamylation is reduced. Furthermore, anterograde IFT and Hedgehog signaling are impaired. This is an interesting approach that will be useful in the field. I have some concerns that need to be addressed before publication.

1. The approach is interesting and will be very useful if it is generally applicable. Can other CCPs or TTLs be recruited to the axoneme to alter tubulin modifications?

2. A major advantage of the assay is the ability to alter tubulin modifications in a rapid and spatially-specific manner. But since the truncated CCP5CD is itself an active enzyme, it can presumably alter glutamylation patterns when expressed in cells. Does it alter the glutamylation of cytoplasmic microtubules? The truncated CCP5CD presumably must cycle through the cilium in order to be trapped by MAP4. Thus, why doesn't it reduce glutamylation of axonemes when expressed, even without the rapamycin treatment? Higher magnification images of the effects of CCP5CD before and after rapamycin treatment need to be shown. Acetylated tubulin is likely not a good marker for other effects of the CCP5 since acetylation is inside the microtubule. What are the effects on deetyrosination?

In figure S6a, truncated CCP5CD does not

Cytosolic glutamylated tubulin can be clearly detected after cold treatment (Supplementary Fig. 8). Expression of CCP5CD did not lead to a noticeable change in tubulin glutamylation compared with the control condition (Neon alone; Supplementary Fig. 8). Recruitment of CCP5CD to the axoneme specifically removed the glutamylation on the axoneme without affecting the level of tubulin glutamylation at other cellular sites including the basal body and cytosol (Supplementary Fig. 8). It is somewhat interesting that expression of soluble CCP5CD in cells had only marginal effect on glutamylation in the cytosol, despite a nature of free diffusion. We speculate that the expression level of cytosolic CCP5CD may be low enough to take an

effect, and/or that endogenous polyglutamylases in the cytosol may counteract CCP5CD-mediated deglutamylation in the cytosol.

We evaluated the effect of deglutamylation on the level of tubulin detyrosination in cilia. Expression of CCP5FL or recruitment of CCP5CD-Neon-FKBP to the axoneme did not significantly affect tubulin detyrosination in cilia, indicating that CCP5-mediated deglutamylation does not affect ciliary detyrosination at least under this condition (Supplementary Fig. 7e,f).

3. Rapamycin-induced anchoring of YFP-FKBP to the axoneme occurred within 2 min (Fig. 1c,d) whereas anchoring of CCP5CD-Neon-FKBP takes much longer (20-30 min?, Fig. 2d). Please provide a time course of the CCP5CD recruitment to the axoneme. Why is the recruitment of CCP5CD so much slower?

We monitored the ciliary entry of CCP5CD-Neon-FKBP by live-cell imaging (Supplementary Fig. 5 and Supplementary Movie 3). The time required for half-maximal accumulation of CCP5CD-Neon-FKBP in the axoneme ($t_{1/2}$) was compared with that of YFP-FKBP (CCP5CD-Neon-FKBP: 95.7 ± 21.9 s vs. YFP-FKBP: 98.1 ± 24.0 s). The reason we incubated cells with rapamycin for 30 min is that CCP5-mediated deglutamylation of purified microtubules requires ~30 min for complete deglutamylation (Berezniuk *et al.*, J Biol Chem, 2013).

4. Does the residual glutamylation at the base of the cilium localize to the centriole and/or at the transition zone?

The residual glutamylation in deglutamylated cilia localized at the inversin zone but not transition zone (Supplementary Fig. 6). The detailed mechanism of how axonemal tubulin in the inversin zone resists the CCP5-mediated deglutamylation is still unclear.

5. The data in Figure 2a-e suggest that glutamylation plays no role in cilia maintenance. However, the suggestion that loss of glutamylation alters anterograde IFT (Figure 3) appears to be at odds with this finding. It seems likely that the time course of the experiment in Figure 2a-e is not sufficient for the authors to conclude anything about the role of glutamylation in cilia maintenance. In particular, work in other systems, particularly *Chlamydomonas*, has shown that it takes many hours of disrupted IFT for the cilia to go away (e.g. Kozminski *et al.* 1995 JCB, Engel *et al.* 2012 JCB; Lin *et al.* 2013 Cilia). Thus, the experiments in Figure 2 need to be repeated with a longer time course of STRIP. In addition, the fact that full length CCP5 localizes to and alters the glutamylation of cilia suggest an easier experiment where expression of CCP5 should prevent ciliation or cause resorption of cilia that are already formed.

Expression of CCP5FL and CCP5CD-cerulean3-MAP4m were used to constitutively deplete tubulin glutamylation (Supplementary Figs. 3 and 4). However, neither of these affected cilia length, indicating that glutamylation is not required for cilia maintenance. Our results demonstrate that the axonemal deglutamylation slows anterograde IFT but does not prohibit it in cilia (Figs. 4, 5 and Supplementary Fig. 11). The Lehtrekk group showed that the demand for anterograde IFT in newly growing cilia is much greater than that in mature cilia (Wren *et al.*, Curr. Biol. 2013). Therefore, the rapid deglutamylation-induced defects we observed in anterograde IFT may cause more drastic inhibition of cilia elongation than of cilia maintenance, a prediction consistent with our finding that rapid deglutamylation impaired elongation of immature, but not mature, cilia (Fig. 3). Long-term deglutamylation does not have enough temporal resolution to specifically uncover its effect on cilia elongation.

6. The data on IFT response to changes in glutamylation are not convincing. I have a hard time seeing any anterograde IFT in Figure 2C so I am not convinced about these data. The anterograde lines that are drawn for the kymograph in Figure 2c suggest that the speed of anterograde IFT is dramatically reduced upon rapamycin treatment but the change displayed in Figure 3d middle does not reflect this. Are the images in Figure 2c representative?

We realize that a single z-stack cannot fully represent the signal of Neon-IFT88. Therefore, in the revised manuscript, we show imaging results by maximal intensity projection (Fig. 4a).

Linescan analysis confirmed that Neon-IFT88 accumulated at the cilia base after STRIP treatment (Fig. 4b).

To improve the quality of the IFT data, we used the software "KymographClear" to process our data (Mangeol *et al*, Mol Biol Cell, 2016). KymographClear greatly improves the quality of IFT kymographs by using distinct colors to highlight the trajectories of anterograde IFT and retrograde IFT (Fig. 4c). Importantly, the improved dataset is fully consistent our original conclusion.

7. The authors demonstrate that reduced glutamylation upon rapamycin treatment alters anterograde IFT (Figure 3), localization of Hedgehog components to the cilium (Figure 4), and transcriptional response to Hedgehog (Figure 4). Based on this, the authors propose that glutamylation alters the motility of IFT motors. An alternative explanation is that rapamycin induced trapping of CCP5CD at MAP4-rich microtubules alters the structure or function of the transition zone and thus the entrance of proteins into the cilium. This could explain the effects on IFT in Figure 3, and trafficking of Hedgehog components in Figure 4. Does the number of anterograde and retrograde IFT events change after rapamycin treatment? It looks like the IFT trains are stuck at the base of the cilium and unable to enter after rapamycin treatment. Does the turnover of Hedgehog pathway components in the cilium change after rapamycin, for example using FRAP analysis of Gli3 at the cilium tip? Does the exit of Hedgehog components Patched or GPR161 change upon rapamycin treatment? If decreased glutamylation is specifically affecting anterograde IFT, then there should be no effect on exit of Patched or GPR161.

Thank you for these very insightful comments. With the processing of KymographClear, we were able to highlight more trajectories of Neon-IFT88 (Fig. 4c). The frequency of Neon-IFT88 particles moving in control and deglutamylated cilia was measured (Supplementary Fig. 11), revealing that deglutamylation only slightly (not statistically significant) decreased the frequency of anterograde IFT and retrograde IFT (Supplementary Fig. 11).

The level of GFP-Gli3 upon stimulation with SAG in deglutamylated cilia was less than that in control cilia (Supplementary Fig. 14b), which is consistent with the finding of the STRIP experiments (Fig. 7d,e). In addition, we performed a FRAP experiment of GFP-Gli3, revealing that axonemal deglutamylation attenuated the ciliary entry of GFP-Gli3 upon stimulation with SAG (Supplementary Fig. 14b), probably owing to defective anterograde IFT.

We also evaluated the redistribution of Patched1-YFP and GPR161 in normal and deglutamylated cilia upon Hh activation (Supplementary Fig. 13), which revealed that axonemal deglutamylation did not affect the ciliary exit of Patched1-YFP and GPR161 (Supplementary Fig. 13).

8. For all graphs, what is being displayed in the y-axis needs to be clarified. For example, for Figure 1f, the figure legend states that fluorescence intensities were measured but the y-axis of the graph indicates average length. Average length of what? Quantification of the changes in both fluorescence intensities and extent of the modifications is critical for the reader to assess the effects of CCP5 recruitment. Why is a normalized cilia length used in Figure 2e rather than the average length as in all the other figures? For all bar graphs, the spread of data across the population needs to be shown using box-and-whisker or dot plots.

We have corrected the information for the y-axis in each of Figs. 1f, 2c, 3c, and 3d, and Supplementary Figs. 1c, 3c, 4c, 7, 8, and 9. For the case of Fig. 2e, to emphasize the change for each cilium at the same scale, the length of each cilium was normalized by dividing by the length value before rapamycin treatment. We have changed our bar graphs concerning cilia length and IFT velocity to box-and-whisker graphs.

Reviewer #3 (Remarks to the Author):

The manuscript by Lin et al aims at studying the role of glutamylation in ciliogenesis, ciliary maintenance, and signaling. The authors devise a clever approach of forced dimerization of a

deglutamyase with an axonemal-targeted MAP. The approach is well characterized using two dimerizers (although the Gibberellin-based system is not used in their experiments later), and the authors find that a limited catalytic domain is captured in the cilia by a FRB-MAP fusion that predominantly localizes to axonemes. The authors have carefully established the validity of their experimental system by demonstrating that (a) other tubulin modifications in axonemes are not impacted, and (b) glutamylation in other cellular locations including cytosol are impacted by the full length CCP5, but not by the isolated catalytic domain. Although the authors have nicely established a novel experimental system for studying glutamylation, which by itself is an important method for the field, I am not convinced by the three conclusions that the authors propose in the current form of the manuscript. I explain reasons for my lack of conviction below:

Thank you for pointing out inadequate nature of our data presentation, which we sincerely dealt with for improvement.

1. Forced CCL5CD localization in cilia prevents ciliogenesis. I am struck by the consistency in cilia length in unstarved unsynchronized cells (<1 μm ; Fig. 2G-H). I am not sure how the authors did the experiment in this case, as 3T3 cells ciliate even in the presence of serum, and/or if they selected for cilia of a certain length.

The authors next measure increase in ciliary length upon starvation for 4h, and notice less increase in case of CCP5CD expression. It is important to emphasize here whether Rapamycin is reversible in its action, so that the dimerization is lost upon changing from Rapamycin containing medium into starvation medium, and that the effects the authors see are because of forced localization of the CCL5CD before initiating starvation. In experiments like these (also in Fig. 4), it is important to show how long it takes for CCP5CD to leave the cilia after removing

Our results revealed that 3T3 cells have various lengths of cilia in unstarved/unsynchronized cells (cells at 0 min of serum starvation; Supplementary Fig. 10). Supplementary Movie 5 also presents two examples of cells having various lengths of cilia upon serum starvation. We carried out an unbiased analysis of cilia length in cells expressing STRIP components. To clearly show the distribution of cilia length, we have changed the bar graphs to box-and-whisker graphs.

The rapamycin-induced dimerization is irreversible (DeRose *et al.*, Pflugers Arch, 2013; Putyrski *et al.*, FEBS Lett, 2012). Previous studies as well as our previous work also showed that the clearance of rapamycin from cells by washout is extremely difficult owing to the high binding affinity between rapamycin and FKBP (dissociation constant = 0.27 nM) (Lin *et al.*, Angew Chem Int Ed Engl, 2013; Hosoi *et al.*, Cancer Res, 1999; Banaszynski *et al.*, J Am Chem Soc, 2005). To make it clearer, we now describe the irreversibility of rapamycin-induced dimerization in the legends of the corresponding figures.

2. Anterograde IFT rates are reduced upon forced CCP5CD localization. I am unable to assess the data provided, given the current quality of movies in supplement, and the gross discrepancy of reported anterograde and retrograde velocities with respect to published literature (for e.g. see PMC4610257, where the anterograde and retrograde IFT velocities in kidney collecting duct cells are close to 1 $\mu\text{m/s}$). The reported anterograde and retrograde IFT velocities here are $\sim 0.5 \mu\text{m/s}$ and $\sim 0.3 \mu\text{m/s}$, respectively, which are very low. The designated lines in kymograph (for e.g., horizontal anterograde lines and thick retrograde lines in Fig. 3C) are most likely not moving particles. I strongly recommend IMCD cells for these experiments, as the cilia are longer than 3T3 cells, and there is available good quality data to compare.

Comparison of the IFT88 dynamics among various cell types revealed that the velocity of each of anterograde and retrograde IFT can vary greatly in anterograde and retrograde IFT (IMCD cells: $0.3\sim 1 \mu\text{m/s}$ for anterograde IFT and $0.6\sim 1 \mu\text{m/s}$ for retrograde IFT; LLC-PK1 cells: $\sim 0.4 \mu\text{m/s}$ for anterograde IFT and $\sim 0.6 \mu\text{m/s}$ for retrograde IFT; MEFs: $\sim 1 \mu\text{m/s}$ for anterograde IFT and $\sim 1 \mu\text{m/s}$ for retrograde IFT; Raman *et al.*, Nat Cell Biol, 2015; He *et al.*, Nat Cell Biol, 2014). This discrepancy in IFT rates may result from distinct PTM patterns and

differences in cilium length among the different cilia model systems (Besschetnova *et al.*, *Curr Biol*, 2010; Gadadhar *et al.*, *J Cell Biol*, 2017; Sirajuddin *et al.*, *Nat Cell Biol*, 2014). In our system, by collecting >850 IFT88 foci from >55 cells, our results revealed that the measured velocities of anterograde IFT and retrograde IFT in NIH3T3 cells were ~0.5 $\mu\text{m/s}$ and ~0.3 $\mu\text{m/s}$, respectively.

To improve the quality of the IFT data, we used the software "KymographClear" to process our data (Mangeol *et al.*, *Mol Biol Cell*, 2016). KymographClear greatly improves the quality of IFT kymographs by using distinct colors to highlight the trajectories of anterograde IFT and retrograde IFT, which enabled us to precisely conduct our kymographic analysis (Fig. 4 and Supplementary Figs. 1, 11, 12).

The cilia of IMCD3 cells are 4- to 5-fold longer than those of NIH3T3 cells. The tubulin PTMs in longer cilia have been confirmed as being different from those of short cilia (i.e., the level of tubulin glycylation increases with cilia length; Gadadhar *et al.*, *J Cell Biol*). Because these factors could also lead to a discrepancy, we were hesitant to use IMCD3 cells in our current study.

3. Reduced Shh signaling upon forced CCP5CD localization. My concerns with this segment of the paper comes from the last panel, where the authors show that increased Gli1 levels in nucleus are reduced upon forced CCP5CD localization in SAG-treated cells. The cilia in these GBS-GFP-3T3 cells are quite long (ranging up to 20 μm , compared to 5 μm in parental cells), and thus given the low increase in Gli1 levels in nucleus in control cells, I am unable to interpret the data. The authors should again discuss the role of reversibility of the dimerization upon SAG treatment, and how long the CCP5CD resides in cilia after Rapamycin is removed.

We apologize for the misleading statement. Indeed, the level of GFP in the nucleus is driven by a promoter with eight Gli binding sites, which serves as a reporter of Gli transcriptional activity. To clarify this statement, we have added a sentence "GFP served as a fluorescence reporter to represent the Gli transcription activity" (Page 19 Line 8) and added "Gli activity" to the figure (Fig. 7). Concerning the long cilia observed for 8XGBS-GFP cells, the cells were starved for 24 h for ciliogenesis and then incubated with rapamycin for 30 min and SAG for another 24 h in serum-free medium. In our pilot experiment with 8XGBS-GFP cells, 2 days of serum starvation increased cilia length (see the figure below).

To mitigate any misleading statement, we now describe the irreversible nature of the CID system, which enabled us to washout excess rapamycin without affecting protein dimerization in the system. This description is made in the legends of the corresponding figures.

4. Finally, an orthogonal approach, such as RNAi for a glutamylase to validate some of the conclusions would strengthen the data.

Two common strategies can be utilized to induce deglutamylation in cells: 1) Depletion or mutation of glutamylases; 2) Overexpression of deglutamylases. The effect of glutamylase depletion on microtubules has been studied, but the conclusions reached based on data from different systems are still controversial (Ikegami *et al.*, 2010; Kubo *et al.*, 2010; Suryavanshi *et al.*, 2010; Pathak *et al.*, 2011; Bosch Grau *et al.*, 2013; Konno *et al.*, 2016; Vogel *et al.*, 2010; Campbel *et al.*, 2002; Regnard *et al.*, 2003; Lee *et al.*, 2013). Alternatively, as in our current study, expression of the deglutamylase CCP5 was used for long-term deglutamylation, which did not affect cilia length, tubulin acetylation, tubulin detyrosination, tubulin polyglutamylation, or tubulin $\Delta 2$ modification. To the least, these results are consistent with the effects induced by STRIP treatment.

Minor comments:

1. Please describe methods in legends to sufficiently understand the figures. For e.g. in Fig. 2 and 4, were the cells confluent before starving?

We have described the methods in the revised figure legends.

2. Cilia lengths vary quite a bit between figures although the authors use 3T3 cells. For e. g cilia lengths are $\sim 10 \mu\text{m}$ (Fig. 1C, 2D), $\sim 5 \mu\text{m}$ (Fig. 1E, 2B), $\sim 20 \mu\text{m}$ (Fig. 4f). Please explain why, and if the different lengths impact on interpretation of data. Please explain the term “normalized cilia length” in Fig. 2E.

Previous studies have confirmed that 5HT₆ expression increases cilia length without noticeably affecting either ciliation efficiency or cilia morphology (Lin *et al.*, Nat Chem Biol, 2013; Su *et al.*, Nat Method, 2013; Phua *et al.*, Cell, 2017). Therefore, cilia length in cells expressing 5HT₆ can reach $\sim 10 \mu\text{m}$ (Figs. 1C, 2D). The length of cilia labeled by glutamylated tubulin is $\sim 2 \mu\text{m}$ (Fig. 2e, f, Supplementary Figs. 1,3,4). The length of Arl13B-labeled cilia is $\sim 2.5 \mu\text{m}$ (Fig. 2b, c, Supplementary Figs. 1,3,4). For the long cilia in 8XGBS-GFP cells, the cells were starved for 24 h for ciliogenesis and then incubated with rapamycin for 30 min and SAG in serum-free medium

for another 24 h. In our pilot experiment with 8XGBS-GFP cells, 2 days of serum starvation increased cilia length (see the figure below).

For the case of Figure 2e, to emphasize the change in each cilium at the same scale, the length of each cilium was normalized by dividing by the length value before rapamycin treatment. In the revised figure legend, we now describe how cilia length was normalized.

3. Velocities in Fig. 3 to be denoted as “ μ m/s” not “s/ μ m”

We have corrected it.

4. Writing needs to be improved throughout the text. Some examples include (a) rewriting parts of the abstract, (b) Page 11 (top line, add “alteration in” cilia length), Page 16 (change “The Lechtreck group once showed”), etc.

Following the reviewer’s suggestion, we had a professional proofreading company thoroughly re-edit the entire manuscript, along with rigorous clarification, and removal of overstated expressions.

5. Methods must include plasmids and/or if P2A-based co-expressing construct was used in all experiments. This is important for data interpretations, as equivalent amounts of FIB-FRB fusions are needed for the experiments.

We now describe the use of P2A-based constructs in the relevant figure legends.

6. Legends should include transfection constructs, cell numbers or confluence before starvation etc for the sake of reproducing the data by other labs.

We now describe the experimental protocols in the relevant figure legends.

7. Methods should mention % of transfected cells in 3T3 cells, and how transfected cells were selected for data collection (in many cases data collection being from <10 total cells from 3-4 experiments).

In most of our experiments, cells needed to be co-transfected with two or three constructs simultaneously. The co-transfection rate in 3T3 cells is 2~5%. Compared with the immunostaining assay, we usually obtained fewer samples in live-cell imaging experiments. The short interval of imaging with multiple channels and z-stacks allowed us to image only 2~3 fields

simultaneously. Under this condition, we usually obtained 8~10 cells from 3~4 independent experiments.

8. Probably, normalization of the fluorescent intensities of CCP5CD-FKBP signal with respect to the MAP-FRB signal is necessary to interpret the data in most figures.

We have now normalized the fluorescence intensities of YFP-FKBP, Neon-mGID1, and CCP5CD-Neon-FKBP by dividing the intensity values by that measured for the corresponding MAP4m (Fig. 1d and Supplementary Fig. 5). Because the value for the MAP4m signal remained essentially consistent in these experiments, the normalized kinetics of FKBP-tagged proteins going onto the axoneme is compared to the value measured by previous method.

We thank the reviewers and the editors for the positive notes and critical suggestions. We have revised the manuscript accordingly and have provided a separate manuscript file with tracked changes. The reviewers' comments are itemized below, followed by our point-by-point responses.

Major changes we have made, including new items:

- 1) Fig. 1d: According to reviewers' suggestion, the intensity measured for YFP-FKBP was divided by that for CFP and plotted.
- 2) Fig. 2c: We have repeated the experiments for measuring cilia length in the three different groups. The results are consistent with our previous finding that deglutamylation does not affect cilia length.
- 3) Fig. 4a: This now shows the maximal intensity projections of Neon-IFT88 before and after rapamycin treatment.
- 4) Fig. 4b: Linescan profiles of Neon-IFT88 in Fig. 4a are now shown.
- 5) Fig. 4c: To improve the quality of the kymograph, time-lapse imaging of Neon-IFT88 in Fig. 4a was utilized to generate kymograph using the software "KymographClear" (see Methods). Red, green, and blue lines represent the trajectories of Neon-IFT88 particles in the anterograde and retrograde directions and static Neon-IFT88, respectively.
- 6) Fig. 4d: Box plots were used to show the velocity of IFT88 under the indicated conditions.
- 7) Fig. 5: We added a FRAP analysis of Neon-IFT88 in control and deglutamylated cilia.
- 8) Fig. 7f: To clearly indicate that the GFP signal represents Gli activity, we have added "(Gli activity)" to the figure.
- 9) Supplementary Fig. 1c: Box plots were used to show the average length of cilia labeled by the indicated antibodies.
- 10) Supplementary Fig. 1d: To improve the quality of the kymograph, the time-lapse imaging of Neon-IFT88 in Supplementary Fig. 1d was utilized to generate kymograph using KymographClear (see Methods). Red, green, and blue lines represent the trajectories of Neon-IFT88 particles in the anterograde and retrograde directions and static Neon-IFT88, respectively.
- 11) Supplementary Fig. 1e: Box plots were used to show the velocity of IFT88 under the indicated conditions.
- 12) Supplementary Fig. 2c: The YFP-FKBP fluorescence intensity was divided by that of CFP and plotted.
- 13) Supplementary Fig. 3c: Box plots were used to show the average length of cilia labeled by the indicated antibodies.
- 14) Supplementary Fig. 4c: Box plots were used to show the average length of cilia labeled by the indicated antibodies.
- 15) Supplementary Fig. 5: The rapamycin-induced translocation of CCP5CD-Neon-FKBP was added.
- 16) Supplementary Fig. 6: We added the distribution of residual glutamylated tubulin after deglutamylation treatment.
- 17) Supplementary Fig. 7b: Box plots were used to show the average length of cilia labeled by the indicated antibodies.
- 18) Supplementary Fig. 7e: The effect of deglutamylation on tubulin deetyrosination was added.
- 19) Supplementary Fig. 7d: Box plots were used to show the average length of cilia labeled by the indicated antibodies.
- 20) Supplementary Fig. 7e: The effect of deglutamylation on the extent of tubulin deetyrosination was added.
- 21) Supplementary Fig. 7f: Box plots were used to show the average length of cilia labeled by the indicated antibodies.
- 22) Supplementary Fig. 8b: Box plots were used to show the average length of glutamylated axoneme and average fluorescence intensity of glutamylated tubulin in the cytosol and basal body.

- 23) Supplementary Fig. 9b: Box plots were used to show the average length of cilia under different conditions.
- 24) Supplementary Fig. 11: We evaluated the effect of long-term deglutamylation on IFT dynamics.
- 25) Supplementary Fig. 12: To improve the quality of the kymograph, time-lapse imaging of Neon-Kif3B in Supplementary Fig. 12a was utilized to generate kymograph using KymographClear (see Methods). Red, green, and blue lines represent the trajectories of Neon-Kif3B particles in the anterograde and retrograde directions and static Neon-Kif3B, respectively.
- 26) Supplementary Fig. 13a,b: We evaluated the effect of deglutamylation on the distribution of patched1 protein.
- 27) Supplementary Fig. 13c,d: We evaluated the effect of deglutamylation on the distribution of GRP161 protein.
- 28) Supplementary Fig. 14: A FRAP analysis of Gli3 in deglutamylation was added.
- 29) Supplementary Movie 3: The translocation of CCP5CD-Neon-FKBP onto the axoneme upon rapamycin treatment was added.
- 30) Supplementary Movie 7: A FRAP analysis of Neon-IFT88 in control and deglutamylation was added.
- 31) Supplementary Movie 9: A FRAP analysis of GFP-Gli3 in control and deglutamylation was added.

Reviewers' comments:

Reviewer #1 (Remarks to the Author):

Comments to the authors:

In this manuscript, the authors have developed a novel methodology to express and direct the deglutamylation CCP5 and its active site domain (CCP5CD) into primary cilia. The concept of using Rapamycin-based dimerization to achieve a precise localization of tubulin-modifying enzymes into cilia is unique and could have a strong impact on future work deciphering the role of a whole range of tubulin PTMs in cilia. Especially the fact that the authors observe almost a complete loss of glutamylation only 30 min after Rapamycin-mediated induction of the system indicates a fast kinetics of PTM modulation, which could previously not be obtained with simply overexpressing modifying enzymes. The beauty of the new system is that tubulin glutamylation can be changed at any point after ciliogenesis, and thus the role of this PTM in ciliogenesis and ciliary functions can be separated, which is what the authors did next.

Using their system, the authors study the role of glutamylation in primary cilia, and confirm the importance of this PTM in cilia biogenesis. Strikingly however they can now also show (using the inducible system) that glutamylation has no impact on the length of cilia once they are assembled. The authors next interrogated the role of polyglutamylation in already assembled cilia, and demonstrated an involvement in Hedgehog signaling: when active CCP5 is localized to cilia, they found an absence of the localization of Hedgehog proteins Smo and Gli as a result of reduced polyglutamylation, indicating a dysfunction of the intra-flagellar transport (IFT).

In summary, the current work is solid and gives new information into the role of tubulin PTMs in primary cilia. However, there are several concerns that need to be addressed before the manuscript can be considered for publication.

We cannot appreciate enough to this reviewer for not only piercing the values of our approach, but also offering constructive expert comments which could greatly help us improve the manuscript. We made our best effort to follow the suggestions.

1. The text contains a number of overstatements and unprecise writing, and thus need rewriting.

The most striking examples are listed below, however the authors should carefully re-read the entire manuscript to avoid unprecise statements.

Examples:

- abstract line 1; page 3 line 5: “A bundle of microtubules called an axoneme...” – a bundle suggests a disordered structure, which is the opposite of what an axoneme is.

We have changed the original sentence “A bundle of microtubules called an axoneme serves as the...” to “The ciliary axoneme serve as the....”. (Page 2, Line 2).

- line 4: “This has led to the concept of a tubulin code,” – overstatement: not the PTMs in the axoneme, but the general diversity of tubulin in all cells of eukaryotic mechanisms led to the formulation of the tubulin code hypothesis.

We have removed the sentence “This has led to the concept of a tubulin code where each unique combination of PMTs...”.

- line 7: “To crack the tubulin code...” – overstatement, the authors just want to alter tubulin glutamylation in cilia, and not crack the entire tubulin code.

We have changed the original sentence “To crack the tubulin code...” to “To address a role of glutamylation in cilia,...” (Page 2, Line 8).

- line 8: “...rewrite tubulin PTMs...” – overstatement: the authors will just remove tubulin glutamylation in the current work.

We have changed the original sentence “...rewrite tubulin PTMs...” to “...deplete tubulin glutamylation...”. (Page 2, Line 8).

- line 9 “... de novo deglutamylation...” – deglutamylation is NOT a modification per se, so it cannot be generated de novo.

We have changed the original sentence “...de novo deglutamylation...” to “...rapid deglutamylation...”. (Page 2, Line 10).

- page 3 line 7: “The C-terminal tails of axonemal tubulin are covalently linked with multiple glutamate side chains through polyglutamylation.” – hard to understand: better state that polyglutamylation generates glutamate chains at the C-terminal tails of tubulin.

We appreciate the suggestion and have changed the original sentence to “polyglutamylation generates glutamate chains at the C-terminal tails of tubulin”. (Page 3, Line 8).

- page 4 line 1: it should be mentioned that some glutamylases also have other substrates than tubulin.

We now describe this as:

...beside tubulin, many nucleocytoplasmic shuttling proteins such as nucleosome assembly protein are also identified as the substrates of glutamylases^{10,35}. Therefore, global manipulation of genes encoding enzymes that modulate glutamylation is insufficient to specifically perturb axonemal glutamylation. (Page 5, Line 14)

Relevant references:

10. van Dijk, J. et al. A Targeted Multienzyme Mechanism for Selective Microtubule Polyglutamylation. *Mol. Cell* 26, 437–448 (2007).
35. Regnard, C. Polyglutamylation of Nucleosome Assembly Proteins. *J. Biol. Chem.* 275, 15969–15976 (2000).

- page 4 line 5: it is true that Berezniuk et al. show that under in vitro conditions, CCP5 is able to shorten long glutamate chains, however this does not exclude that this activity is not predominant in vivo. This should be discussed in the light of the following publication, which shows that this is indeed the case: Wu H-Y, Wei P, Morgan JI (2017) Role of Cytosolic Carboxypeptidase 5 in Neuronal Survival and Spermatogenesis. *Sci Rep* 7: 41428

We have added the reference “Wu *et al.* (2017)” at the end of the sentence “CCP5 preferentially removes a glutamate at the branching fork, whereas other CCP members target a glutamate residue in a linear, tandem sequence in vivo”. (Page 4, Line 5)

Relevant reference:

13. Wu, H.-Y., Wei, P. & Morgan, J. I. Role of Cytosolic Carboxypeptidase 5 in Neuronal Survival and Spermatogenesis. *Sci. Rep.* 7, 41428 (2017)

- page 4 line 12: The authors provide a list of how tubulin glutamylation was studied so far, and completely forgot the multiple publications on TLL- and CCP-KO animals that have been published over the last years.

We have added the relevant references as below.

Relevant references:

19. Kubo, T., aki Yanagisawa, H., Yagi, T., Hirono, M. & Kamiya, R. Tubulin Polyglutamylation Regulates Axonemal Motility by Modulating Activities of Inner-Arm Dyneins. *Curr. Biol.* 20, 441–445 (2010).
20. Pathak, N., Austin, C. A. & Drummond, I. A. Tubulin tyrosine ligase-like genes *tll3* and *tll6* maintain zebrafish cilia structure and motility. *J. Biol. Chem.* 286, 11685–11695 (2011).
21. Chen, D. et al. The zebrafish *flee* gene encodes an essential regulator of cilia tubulin polyglutamylation. *Mol. Biol. Cell* 19, 308–317 (2007).
22. Lyons, P. J., Sapio, M. R. & Fricker, L. D. Zebrafish cytosolic carboxypeptidases 1 and 5 are essential for embryonic development. *J. Biol. Chem.* 288, 30454–30462 (2013).
23. Suryavanshi, S. et al. Tubulin Glutamylation Regulates Ciliary Motility by Altering Inner Dynein Arm Activity. *Curr. Biol.* 20, 435–440 (2010).
24. Ikegami, K., Sato, S., Nakamura, K., Ostrowski, L. E. & Setou, M. Tubulin polyglutamylation is essential for airway ciliary function through the regulation of beating asymmetry. *Proc. Natl. Acad. Sci. U. S. A.* 107, 10490–10495 (2010).
25. Lee, J. E. et al. CEP41 is mutated in Joubert syndrome and is required for tubulin glutamylation at the cilium. *Nat. Genet.* 44, 193–199 (2012).
26. Alford, L. M. et al. The nexin link and B-tubule glutamylation maintain the alignment of outer doublets in the ciliary axoneme. *Cytoskeleton* 73, 331–340 (2016).
27. Grau, M. B. et al. Tubulin glycolases and glutamylases have distinct functions in stabilization and motility of ependymal cilia. *J. Cell Biol.* 202, 441–451 (2013).
28. Kim, J. et al. Functional genomic screen for modulators of ciliogenesis and cilium length. *Nature* 464, 1048–1051 (2010).
29. Konno A, Ikegami K, Konishi Y, Yang HJ, Abe M, Yamazaki M, Sakimura K, Yao I, Shiba K, I. K. and S. M. Doublet 7 shortening, doublet 5-preferential poly-Glu reduction, and beating stall of sperm flagella in *Tll9^{-/-}* mice. *J. Cell Sci.* pii:jcs.185983 (2016).
30. Kubo, T., Yagi, T. & Kamiya, R. Tubulin polyglutamylation regulates flagellar motility by controlling a specific inner-arm dynein that interacts with the dynein regulatory complex. *Cytoskeleton* 69, 1059–1068 (2012).
31. Lee, G. S. et al. Disruption of *Tll5/Stamp* gene (tubulin tyrosine ligase-like protein 5/SRC-1 and TIF2-associated modulatory protein gene) in male mice causes sperm malformation and infertility. *J. Biol. Chem.* 288, 15167–15180 (2013).
32. O’Hagan, R. et al. The tubulin deglutamylase CCPP-1 regulates the function and stability of sensory cilia in *C. elegans*. *Curr. Biol.* 21, 1685–1694 (2011).

33. Vogel, P., Hansen, G., Fontenot, G. & Read, R. Tubulin tyrosine ligase-like 1 deficiency results in chronic rhinosinusitis and abnormal development of spermatid flagella in mice. *Vet. Pathol.* 47, 703–12 (2010).

- page 4 line 16: spastin is not recruited to microtubules by polyglutamylation, but it is activated on the microtubules. Spastin alone can localize to microtubules via a specific microtubule binding domain. This has been shown in the ref 15 which the authors cite. They should additionally cite Valenstein ML, Roll-Mecak A (2016) Graded Control of Microtubule Severing by Tubulin Glutamylation. *Cell* 164: 911-921.

We have added the reference and corrected the original sentence “the overexpression of the polyglutamylase TLL6 in HeLa cells leads to microtubule disassembly due to the recruitment of a severing enzyme, namely spastin,...” to “Tubulin hyperglutamylation leads to microtubule disassembly owing to the binding of a severing enzyme...”. (Page 4, Line 14)

Reference:

17. Valenstein, M. L., Roll-mecak, A., Valenstein, M. L. & Roll-mecak, A. Graded Control of Microtubule Severing by Tubulin Glutamylation. *Cell* 164, 911–921 (2016).

- page 14 line 4: “Polyglutamylation on the surface of axonemes serves as an interface between microtubules and IFT motors.” – overstatement: this has never been shown. What has been shown is that polyglutamylation somehow alters certain interactions between IFT motors and axonemal microtubules. The interface of many microtubule motors is NOT the C-terminal tail, where glutamylation takes place.

We have changed the original sentence “Polyglutamylation on the surface of axoneme serves as an interface between microtubules and IFT motors, suggesting the possibility that...” to “Besides mechanical support, axoneme also serve as railways for IFT. We thus evaluated whether axonemal deglutamylation impacts the rate of IFT”. (Page 14, Line 15)

2. The authors have constructed a set of constructs derived from the deglutamylase CCP5, which they attract to the cilia by dimerizing them with the FRB/FKRB rapamycin system with the microtubule-associated protein MAP4. This system should allow to focus CCP5 into the cilia and thus lead to a specific deglutamylation only of the ciliary microtubules – the axonemes. How do the authors make sure that the active CCP5 constructs do not deglutamylate cellular substrates such as microtubules and other glutamylated proteins BEFORE they localize the enzyme to cilia? Obviously in standard cultured cell lines, there is little glutamylation on interphase microtubules, but there are ways of visualizing even weak glutamylation levels by using elevated concentrations of anti-glutamylation antibodies (see Magiera MM, Janke C (2013) Investigating tubulin posttranslational modifications with specific antibodies. In *Methods Cell Biol*, Correia JJ, Wilson L (eds), Vol. 115, 2013/08/27 edn, pp 247-267. Burlington: Academic Press).

Cytosolic glutamylated tubulin can be clearly detected after cold treatment (Supplementary Fig. 8). Expression of CCP5CD did not lead to a noticeable change in tubulin glutamylation compared with the control condition (Neon alone; Supplementary Fig. 8). Recruitment of CCP5CD to the axoneme specifically removed the glutamylation on the axoneme without affecting the level of tubulin glutamylation at other cellular sites including the basal body and cytosol (Supplementary Fig. 8). It is somewhat interesting that expression of soluble CCP5CD in cells had only marginal effect on glutamylation in the cytosol, despite a nature of free diffusion. We speculate that the expression level of cytosolic CCP5CD may be low enough to take an effect, and/or that endogenous polyglutamylases in the cytosol may counteract CCP5CD-mediated deglutamylation in the cytosol.

3. In Fig. 1e, the authors show that 30 min after induction of ciliary deglutamylation with Rapamycin, the CCP5CD-Neon-FKBP recruited to the cilia is able to deglutamylate almost all of the glutamylated tubulin, except the proximal end of the cilia. The rapamycin, they still residual

glutamylation at the base of the cilia. They suggest that this could be due to the activity of endogenous TLL glutamylases present at the base of the cilia. However, considering that CCP5 is overexpressed as compared to the endogenous TLLs, why do the authors think that CCP5 could not overcome this supposedly much lower endogenous activity? They should check the glutamylation status of the ciliary base at later time points after CCP5 induction to verify if this glutamylation remains.

To induce long-term deglutamylation in cilia, we forcefully anchored the CCP5 catalytic domain (CCP5CD) to the axoneme via fusion with MAP4m. Accumulation of CCP5CD-MAP4m in cilia constitutively eliminated axonemal glutamylation, which still could not remove the residual glutamylation at the proximal end of the cilia (Supplementary Fig. 4b). The residual glutamylation localized to the inversin zone but not the transition zone (Supplementary Fig. 6). The detailed mechanism of how axonemal tubulins in the inversin zone resist deglutamylation treatment is unclear.

4. In Fig. 2, the authors aim to show that STRIP does not alter the average length of the cilium. However their quantification shows that the average length in the CCP5CD-Neon-FKBP cells is 2 μm , while it is 2.5 μm in the Neon-only cells. This represents a 20% change in ciliary length, and it might turn out significant if the authors would measure more cilia. To clarify this point, a more thorough analysis (measuring more cilia, proper statistic analyses) is required to elucidate the role of CCP5 activity, and thus glutamylation, in ciliary length control.

We conducted two more independent experiments for cilia length measurement. The results show that the cilia length of cells expressing CCP5CD-Neon-FKBP was comparable with that of control cells after STRIP treatment ($2.33 \pm 0.19 \mu\text{m}$ in Neon-FKBP group vs. $2.39 \pm 0.16 \mu\text{m}$ in CCP5CD-Neon-FKBP group; $P = 0.81$; Fig. 2c).

5. At page 13, the authors write that the steady-state of primary cilia structure is independent of the glutamylation state. As it has already been established that the other tubulin posttranslational modification, glycylation, is linked to glutamylation and plays a role in stabilizing cilia (Bosch Grau, M et al. 2013. Tubulin glycylation and glutamylases have distinct functions in stabilization and motility of ependymal cilia. *J Cell Bio* 202: 441–45; Bosch Grau M et al. (2017) Alterations in the balance of tubulin glycylation and glutamylation in photoreceptors leads to retinal degeneration. *J Cell Sci* 130: 938-949), it will be necessary to check if glycylation is altered (increased?) in the presence of CCP5CD-Neon-FKBP.

We have used two commercially available antibodies against tubulin glycylation, namely anti-polyglycylated tubulin (AXO49, MABS276, Millipore) and anti-monoglycylated tubulin (TAP952, MABS277, Sigma-Aldrich), together with six different fixation protocols to label tubulin glycylation in primary cilia in NIH3T3 cells. Unfortunately, none of them could detect tubulin glycylation in primary cilia. Tubulin glycylation has been exclusively detected in motile cilia and flagella by using anti-TAP952 (Redeker *et al.*, *Science*, 1994; Rudiger *et al.*, *FEBS Lett*, 1995; Bre *et al.*, *Mol Biol Cell*, 1996; Weber *et al.*, *FEBS Lett*, 1996; Xia *et al.*, *J Cell Biol*, 2000). In the case of primary cilia, however, Dr. Janke and colleagues confirmed that anti-TAP952 failed to detect cilia glycylation in fibroblasts and other cell types (Rocha *et al.*, *EMBO J*, 2014; Gadadhar *et al.*, *J Cell Biol*, 2017). Very recently, the same group produced new glycylation-specific antibodies that enabled a clear detection of glycylation in very long primary cilia ($>10 \mu\text{m}$) but not (or rarely) in short primary cilia ($<9 \mu\text{m}$) (Gadadhar *et al.*, *J. Cell Biol*, 2017). Because the average length of primary cilia in NIH3T3 cells is usually $<5 \mu\text{m}$, it was challenging to detect tubulin glycylation in our system.

6. In Fig. 3 the authors show that rapid tubulin deglutamylation resulting in loss of signal for Neon-IFT88 on the axonemes. However, when they measure the anterograde movement of Neon-IFT88, the reduction of IFT88 mobility is not very drastic: they determine a 0.15 sec/ μm slower transport in the absence of glutamylation. This stands in strong contrast to their immunofluorescence images, in which hardly any IFT88 is observed in cilia, and where IFT88 accumulates at the basal bodies. These two results are somewhat contradictory, and raise the

question if the authors have looked into the IFT88 movement after a longer period of time after Rapamycin induction. Moreover, the authors should check if IFT88 does not enter the cilium at all, and if this is the case, if there is a reduced rate of ciliary entry that could explain the quasi-absence of IFT88 from deglutamylated cilia.

We realize that a single z-stack cannot fully represent the signal of Neon-IFT88. Therefore, in the revised manuscript, we show the imaging results by maximal intensity projection (Fig. 4a). Linescan analysis confirmed that Neon-IFT88 accumulates at the cilia base after STRIP treatment (Fig. 4b).

To study the effect of long-term deglutamylation on IFT, we measured the motility of Neon-IFT88 in cilia expressing CCP5CD-mCh-MAP4m (Supplementary Fig. 11). The results revealed that deglutamylation triggered by CCP5CD-mCh-MAPm slows anterograde IFT with no obviously effect on retrograde IFT, a result that is similar to the case for STRIP-induced rapid deglutamylation (Supplementary Fig. 11). Therefore, long-term deglutamylation does not have a more drastic effect on anterograde IFT compared with short-term treatment.

We used FRAP to check the ciliary entry and dynamics of IFT88 in normal and deglutamylated cilia. The results demonstrated that long-term deglutamylation slows the recovery of Neon-IFT88 in the axoneme without affecting the tethering of IFT88 to the basal body (Fig. 5). This suggests that deglutamylation decreases the motility of Neon-IFT88 along the axoneme in the anterograde direction. Together with kymographic results acquired for the slow anterograde IFT in deglutamylated cilia (Fig. 4d and Supplementary Fig. 11), we demonstrate that axonemal deglutamylation slows down anterograde IFT.

7. In Fig. 3g, the authors show a redistribution of Kif3B between the base and the rest of the cilia after deglutamylation. They show an enrichment of Kif3B from about 60% in control to about 70% in deglutamylated cilia, and state: "In summary, these results support that glutamylation controls anterograde IFT through Kif3B of the kinesin-2 complex." (intersection of pages 15 & 16). Considering the small change in localization, they might want to discuss why they think there is only such a subtle effect, considering their strong claim that the mechanism of control is a regulation of the Kif3b motor.

They in fact discuss this in the discussion (intersection of pages 18 and 19: "Another explanation is that residual anterograde IFT in the STRIP cells may have been sufficient for cilia growth and maintenance, unlike the cases for near complete loss-of-function of kinesin-2 proteins". Thus there is no need of such a strong conclusion in the result section

We have changed the original sentence "In summary, these results support that glutamylation controls anterograde IFT through Kif3B of the kinesin-2 complex" to "In summary, these results suggest that glutamylation controls anterograde IFT through Kif3B of the kinesin2-complex". (Page17 Line12)

8. In Fig. 4 the authors show that the localization of the signaling components Smo3 and Gli is impaired when axonemes are deglutamylated by the CCP5CD-Neon-FKBP upon Rapamycin induction. They also show that Gli is localized towards the tip of the cilium, whereas Smo localizes throughout the cilium. How do the authors explain the loss of the Gli signal from the tip of the cilium upon deglutamylation?

The kinesin-2 complex directly interacts with Hedgehog components such as Gli2 and Gli3 (Carpenter *et al.*, J Cell Sci, 2015). Moreover, knockout of IFT25, a component of the IFT-B complex, inhibits the trafficking of Gli to the cilia tip from the cytosol upon treatment of cells with Hedgehog ligand (Keady *et al.*, Dev Cell, 2012). These studies suggest that the anterograde IFT complex binds Hedgehog components and transports them to cilia where they regulate Hedgehog signaling. We therefore hypothesized that a deglutamylation-induced defect in anterograde IFT would affect the redistribution of Gli from the cytosol to the ciliary tip upon Hedgehog activation. Indeed, Our FRAP results confirmed that ciliary entry of Gli upon Hedgehog activation was impaired by deglutamylation (Supplementary Fig. 14), likely through an anterograde IFT-dependent mechanism (Fig. 7d).

Reviewer #2 (Remarks to the Author):

The manuscript by Wang et al describes a new approach to altering tubulin modifications in cells in a controlled manner. The authors utilize rapamycin-induced recruitment of a deglutamylase, CCP5, to the axoneme and show that glutamylation is reduced. Furthermore, anterograde IFT and Hedgehog signaling are impaired. This is an interesting approach that will be useful in the field. I have some concerns that need to be addressed before publication.

1. The approach is interesting and will be very useful if it is generally applicable. Can other CCPs or TTLs be recruited to the axoneme to alter tubulin modifications?

According to our previous study, almost all soluble forms of PTM modifying enzymes should be able to translocate to the axoneme by dimerizer treatment (Lin *et al.*, Nat Chem Biol, 2013). In our pilot experiment, we successfully translocated the YFP-FKBP-fused catalytic domain of TTL6 (YFP-FKBP-TTL6CD) to the ciliary axoneme (see results below). For the current study, however, we prefer to focus on the effect of inducible axonemal deglutamylation on ciliary structure and functions; in future projects, we plan to apply this established tool to comprehensively study other tubulin PTMs.

2. A major advantage of the assay is the ability to alter tubulin modifications in a rapid and spatially-specific manner. But since the truncated CCP5CD is itself an active enzyme, it can presumably alter glutamylation patterns when expressed in cells. Does it alter the glutamylation of cytoplasmic microtubules? The truncated CCP5CD presumably must cycle through the cilium in order to be trapped by MAP4. Thus, why doesn't it reduce glutamylation of axonemes when expressed, even without the rapamycin treatment? Higher magnification images of the effects of CCP5CD before and after rapamycin treatment need to be shown. Acetylated tubulin is likely not a good marker for other effects of the CCP5 since acetylation is inside the microtubule. What are the effects on detyrosination?

In figure S6a, truncated CCP5CD does not

Cytosolic glutamylated tubulin can be clearly detected after cold treatment (Supplementary Fig. 8). Expression of CCP5CD did not lead to a noticeable change in tubulin glutamylation compared with the control condition (Neon alone; Supplementary Fig. 8). Recruitment of CCP5CD to the axoneme specifically removed the glutamylation on the axoneme without affecting the level of tubulin glutamylation at other cellular sites including the basal body and cytosol (Supplementary Fig. 8). It is somewhat interesting that expression of soluble CCP5CD in cells had only marginal effect on glutamylation in the cytosol, despite a nature of free diffusion. We speculate that the expression level of cytosolic CCP5CD may be low enough to take an

effect, and/or that endogenous polyglutamylases in the cytosol may counteract CCP5CD-mediated deglutamylation in the cytosol.

We evaluated the effect of deglutamylation on the level of tubulin detyrosination in cilia. Expression of CCP5FL or recruitment of CCP5CD-Neon-FKBP to the axoneme did not significantly affect tubulin detyrosination in cilia, indicating that CCP5-mediated deglutamylation does not affect ciliary detyrosination at least under this condition (Supplementary Fig. 7e,f).

3. Rapamycin-induced anchoring of YFP-FKBP to the axoneme occurred within 2 min (Fig. 1c,d) whereas anchoring of CCP5CD-Neon-FKBP takes much longer (20-30 min?, Fig. 2d). Please provide a time course of the CCP5CD recruitment to the axoneme. Why is the recruitment of CCP5CD so much slower?

We monitored the ciliary entry of CCP5CD-Neon-FKBP by live-cell imaging (Supplementary Fig. 5 and Supplementary Movie 3). The time required for half-maximal accumulation of CCP5CD-Neon-FKBP in the axoneme ($t_{1/2}$) was compared with that of YFP-FKBP (CCP5CD-Neon-FKBP: 95.7 ± 21.9 s vs. YFP-FKBP: 98.1 ± 24.0 s). The reason we incubated cells with rapamycin for 30 min is that CCP5-mediated deglutamylation of purified microtubules requires ~30 min for complete deglutamylation (Berezniuk *et al.*, J Biol Chem, 2013).

4. Does the residual glutamylation at the base of the cilium localize to the centriole and/or at the transition zone?

The residual glutamylation in deglutamylated cilia localized at the inversin zone but not transition zone (Supplementary Fig. 6). The detailed mechanism of how axonemal tubulin in the inversin zone resists the CCP5-mediated deglutamylation is still unclear.

5. The data in Figure 2a-e suggest that glutamylation plays no role in cilia maintenance. However, the suggestion that loss of glutamylation alters anterograde IFT (Figure 3) appears to be at odds with this finding. It seems likely that the time course of the experiment in Figure 2a-e is not sufficient for the authors to conclude anything about the role of glutamylation in cilia maintenance. In particular, work in other systems, particularly *Chlamydomonas*, has shown that it takes many hours of disrupted IFT for the cilia to go away (e.g. Kozminski *et al.* 1995 JCB, Engel *et al.* 2012 JCB; Lin *et al.* 2013 Cilia). Thus, the experiments in Figure 2 need to be repeated with a longer time course of STRIP. In addition, the fact that full length CCP5 localizes to and alters the glutamylation of cilia suggest an easier experiment where expression of CCP5 should prevent ciliation or cause resorption of cilia that are already formed.

Expression of CCP5FL and CCP5CD-cerulean3-MAP4m were used to constitutively deplete tubulin glutamylation (Supplementary Figs. 3 and 4). However, neither of these affected cilia length, indicating that glutamylation is not required for cilia maintenance. Our results demonstrate that the axonemal deglutamylation slows anterograde IFT but does not prohibit it in cilia (Figs. 4, 5 and Supplementary Fig. 11). The Lechtreck group showed that the demand for anterograde IFT in newly growing cilia is much greater than that in mature cilia (Wren *et al.*, Curr. Biol. 2013). Therefore, the rapid deglutamylation-induced defects we observed in anterograde IFT may cause more drastic inhibition of cilia elongation than of cilia maintenance, a prediction consistent with our finding that rapid deglutamylation impaired elongation of immature, but not mature, cilia (Fig. 3). Long-term deglutamylation does not have enough temporal resolution to specifically uncover its effect on cilia elongation.

6. The data on IFT response to changes in glutamylation are not convincing. I have a hard time seeing any anterograde IFT in Figure 2C so I am not convinced about these data. The anterograde lines that are drawn for the kymograph in Figure 2c suggest that the speed of anterograde IFT is dramatically reduced upon rapamycin treatment but the change displayed in Figure 3d middle does not reflect this. Are the images in Figure 2c representative?

We realize that a single z-stack cannot fully represent the signal of Neon-IFT88. Therefore, in the revised manuscript, we show imaging results by maximal intensity projection (Fig. 4a).

Linescan analysis confirmed that Neon-IFT88 accumulated at the cilia base after STRIP treatment (Fig. 4b).

To improve the quality of the IFT data, we used the software "KymographClear" to process our data (Mangeol *et al*, Mol Biol Cell, 2016). KymographClear greatly improves the quality of IFT kymographs by using distinct colors to highlight the trajectories of anterograde IFT and retrograde IFT (Fig. 4c). Importantly, the improved dataset is fully consistent our original conclusion.

7. The authors demonstrate that reduced glutamylation upon rapamycin treatment alters anterograde IFT (Figure 3), localization of Hedgehog components to the cilium (Figure 4), and transcriptional response to Hedgehog (Figure 4). Based on this, the authors propose that glutamylation alters the motility of IFT motors. An alternative explanation is that rapamycin induced trapping of CCP5CD at MAP4-rich microtubules alters the structure or function of the transition zone and thus the entrance of proteins into the cilium. This could explain the effects on IFT in Figure 3, and trafficking of Hedgehog components in Figure 4. Does the number of anterograde and retrograde IFT events change after rapamycin treatment? It looks like the IFT trains are stuck at the base of the cilium and unable to enter after rapamycin treatment. Does the turnover of Hedgehog pathway components in the cilium change after rapamycin, for example using FRAP analysis of Gli3 at the cilium tip? Does the exit of Hedgehog components Patched or GPR161 change upon rapamycin treatment? If decreased glutamylation is specifically affecting anterograde IFT, then there should be no effect on exit of Patched or GPR161.

Thank you for these very insightful comments. With the processing of KymographClear, we were able to highlight more trajectories of Neon-IFT88 (Fig. 4c). The frequency of Neon-IFT88 particles moving in control and deglutamylated cilia was measured (Supplementary Fig. 11), revealing that deglutamylation only slightly (not statistically significant) decreased the frequency of anterograde IFT and retrograde IFT (Supplementary Fig. 11).

The level of GFP-Gli3 upon stimulation with SAG in deglutamylated cilia was less than that in control cilia (Supplementary Fig. 14b), which is consistent with the finding of the STRIP experiments (Fig. 7d,e). In addition, we performed a FRAP experiment of GFP-Gli3, revealing that axonemal deglutamylation attenuated the ciliary entry of GFP-Gli3 upon stimulation with SAG (Supplementary Fig. 14b), probably owing to defective anterograde IFT.

We also evaluated the redistribution of Patched1-YFP and GPR161 in normal and deglutamylated cilia upon Hh activation (Supplementary Fig. 13), which revealed that axonemal deglutamylation did not affect the ciliary exit of Patched1-YFP and GPR161 (Supplementary Fig. 13).

8. For all graphs, what is being displayed in the y-axis needs to be clarified. For example, for Figure 1f, the figure legend states that fluorescence intensities were measured but the y-axis of the graph indicates average length. Average length of what? Quantification of the changes in both fluorescence intensities and extent of the modifications is critical for the reader to assess the effects of CCP5 recruitment. Why is a normalized cilia length used in Figure 2e rather than the average length as in all the other figures? For all bar graphs, the spread of data across the population needs to be shown using box-and-whisker or dot plots.

We have corrected the information for the y-axis in each of Figs. 1f, 2c, 3c, and 3d, and Supplementary Figs. 1c, 3c, 4c, 7, 8, and 9. For the case of Fig. 2e, to emphasize the change for each cilium at the same scale, the length of each cilium was normalized by dividing by the length value before rapamycin treatment. We have changed our bar graphs concerning cilia length and IFT velocity to box-and-whisker graphs.

Reviewer #3 (Remarks to the Author):

The manuscript by Lin et al aims at studying the role of glutamylation in ciliogenesis, ciliary maintenance, and signaling. The authors devise a clever approach of forced dimerization of a

deglutamyase with an axonemal-targeted MAP. The approach is well characterized using two dimerizers (although the Gibberellin-based system is not used in their experiments later), and the authors find that a limited catalytic domain is captured in the cilia by a FRB-MAP fusion that predominantly localizes to axonemes. The authors have carefully established the validity of their experimental system by demonstrating that (a) other tubulin modifications in axonemes are not impacted, and (b) glutamylation in other cellular locations including cytosol are impacted by the full length CCP5, but not by the isolated catalytic domain. Although the authors have nicely established a novel experimental system for studying glutamylation, which by itself is an important method for the field, I am not convinced by the three conclusions that the authors propose in the current form of the manuscript. I explain reasons for my lack of conviction below:

Thank you for pointing out inadequate nature of our data presentation, which we sincerely dealt with for improvement.

1. Forced CCL5CD localization in cilia prevents ciliogenesis. I am struck by the consistency in cilia length in unstarved unsynchronized cells (<1 μm ; Fig. 2G-H). I am not sure how the authors did the experiment in this case, as 3T3 cells ciliate even in the presence of serum, and/or if they selected for cilia of a certain length.

The authors next measure increase in ciliary length upon starvation for 4h, and notice less increase in case of CCP5CD expression. It is important to emphasize here whether Rapamycin is reversible in its action, so that the dimerization is lost upon changing from Rapamycin containing medium into starvation medium, and that the effects the authors see are because of forced localization of the CCL5CD before initiating starvation. In experiments like these (also in Fig. 4), it is important to show how long it takes for CCP5CD to leave the cilia after removing

Our results revealed that 3T3 cells have various lengths of cilia in unstarved/unsynchronized cells (cells at 0 min of serum starvation; Supplementary Fig. 10). Supplementary Movie 5 also presents two examples of cells having various lengths of cilia upon serum starvation. We carried out an unbiased analysis of cilia length in cells expressing STRIP components. To clearly show the distribution of cilia length, we have changed the bar graphs to box-and-whisker graphs.

The rapamycin-induced dimerization is irreversible (DeRose *et al.*, Pflugers Arch, 2013; Putyrski *et al.*, FEBS Lett, 2012). Previous studies as well as our previous work also showed that the clearance of rapamycin from cells by washout is extremely difficult owing to the high binding affinity between rapamycin and FKBP (dissociation constant = 0.27 nM) (Lin *et al.*, Angew Chem Int Ed Engl, 2013; Hosoi *et al.*, Cancer Res, 1999; Banaszynski *et al.*, J Am Chem Soc, 2005). To make it clearer, we now describe the irreversibility of rapamycin-induced dimerization in the legends of the corresponding figures.

2. Anterograde IFT rates are reduced upon forced CCP5CD localization. I am unable to assess the data provided, given the current quality of movies in supplement, and the gross discrepancy of reported anterograde and retrograde velocities with respect to published literature (for e.g. see PMC4610257, where the anterograde and retrograde IFT velocities in kidney collecting duct cells are close to 1 $\mu\text{m/s}$). The reported anterograde and retrograde IFT velocities here are $\sim 0.5 \mu\text{m/s}$ and $\sim 0.3 \mu\text{m/s}$, respectively, which are very low. The designated lines in kymograph (for e.g., horizontal anterograde lines and thick retrograde lines in Fig. 3C) are most likely not moving particles. I strongly recommend IMCD cells for these experiments, as the cilia are longer than 3T3 cells, and there is available good quality data to compare.

Comparison of the IFT88 dynamics among various cell types revealed that the velocity of each of anterograde and retrograde IFT can vary greatly in anterograde and retrograde IFT (IMCD cells: $0.3\sim 1 \mu\text{m/s}$ for anterograde IFT and $0.6\sim 1 \mu\text{m/s}$ for retrograde IFT; LLC-PK1 cells: $\sim 0.4 \mu\text{m/s}$ for anterograde IFT and $\sim 0.6 \mu\text{m/s}$ for retrograde IFT; MEFs: $\sim 1 \mu\text{m/s}$ for anterograde IFT and $\sim 1 \mu\text{m/s}$ for retrograde IFT; Raman *et al.*, Nat Cell Biol, 2015; He *et al.*, Nat Cell Biol, 2014). This discrepancy in IFT rates may result from distinct PTM patterns and

differences in cilium length among the different cilia model systems (Besschetnova *et al.*, *Curr Biol*, 2010; Gadadhar *et al.*, *J Cell Biol*, 2017; Sirajuddin *et al.*, *Nat Cell Biol*, 2014). In our system, by collecting >850 IFT88 foci from >55 cells, our results revealed that the measured velocities of anterograde IFT and retrograde IFT in NIH3T3 cells were ~0.5 $\mu\text{m/s}$ and ~0.3 $\mu\text{m/s}$, respectively.

To improve the quality of the IFT data, we used the software "KymographClear" to process our data (Mangeol *et al.*, *Mol Biol Cell*, 2016). KymographClear greatly improves the quality of IFT kymographs by using distinct colors to highlight the trajectories of anterograde IFT and retrograde IFT, which enabled us to precisely conduct our kymographic analysis (Fig. 4 and Supplementary Figs. 1, 11, 12).

The cilia of IMCD3 cells are 4- to 5-fold longer than those of NIH3T3 cells. The tubulin PTMs in longer cilia have been confirmed as being different from those of short cilia (i.e., the level of tubulin glycylation increases with cilia length; Gadadhar *et al.*, *J Cell Biol*). Because these factors could also lead to a discrepancy, we were hesitant to use IMCD3 cells in our current study.

3. Reduced Shh signaling upon forced CCP5CD localization. My concerns with this segment of the paper comes from the last panel, where the authors show that increased Gli1 levels in nucleus are reduced upon forced CCP5CD localization in SAG-treated cells. The cilia in these GBS-GFP-3T3 cells are quite long (ranging up to 20 μm , compared to 5 μm in parental cells), and thus given the low increase in Gli1 levels in nucleus in control cells, I am unable to interpret the data. The authors should again discuss the role of reversibility of the dimerization upon SAG treatment, and how long the CCP5CD resides in cilia after Rapamycin is removed.

We apologize for the misleading statement. Indeed, the level of GFP in the nucleus is driven by a promoter with eight Gli binding sites, which serves as a reporter of Gli transcriptional activity. To clarify this statement, we have added a sentence "GFP served as a fluorescence reporter to represent the Gli transcription activity" (Page 19 Line 8) and added "Gli activity" to the figure (Fig. 7). Concerning the long cilia observed for 8XGBS-GFP cells, the cells were starved for 24 h for ciliogenesis and then incubated with rapamycin for 30 min and SAG for another 24 h in serum-free medium. In our pilot experiment with 8XGBS-GFP cells, 2 days of serum starvation increased cilia length (see the figure below).

To mitigate any misleading statement, we now describe the irreversible nature of the CID system, which enabled us to washout excess rapamycin without affecting protein dimerization in the system. This description is made in the legends of the corresponding figures.

4. Finally, an orthogonal approach, such as RNAi for a glutamylase to validate some of the conclusions would strengthen the data.

Two common strategies can be utilized to induce deglutamylation in cells: 1) Depletion or mutation of glutamylases; 2) Overexpression of deglutamylases. The effect of glutamylase depletion on microtubules has been studied, but the conclusions reached based on data from different systems are still controversial (Ikegami *et al.*, 2010; Kubo *et al.*, 2010; Suryavanshi *et al.*, 2010; Pathak *et al.*, 2011; Bosch Grau *et al.*, 2013; Konno *et al.*, 2016; Vogel *et al.*, 2010; Campbel *et al.*, 2002; Regnard *et al.*, 2003; Lee *et al.*, 2013). Alternatively, as in our current study, expression of the deglutamylase CCP5 was used for long-term deglutamylation, which did not affect cilia length, tubulin acetylation, tubulin detyrosination, tubulin polyglutamylation, or tubulin $\Delta 2$ modification. To the least, these results are consistent with the effects induced by STRIP treatment.

Minor comments:

1. Please describe methods in legends to sufficiently understand the figures. For e.g. in Fig. 2 and 4, were the cells confluent before starving?

We have described the methods in the revised figure legends.

2. Cilia lengths vary quite a bit between figures although the authors use 3T3 cells. For e. g cilia lengths are $\sim 10 \mu\text{m}$ (Fig. 1C, 2D), $\sim 5 \mu\text{m}$ (Fig. 1E, 2B), $\sim 20 \mu\text{m}$ (Fig. 4f). Please explain why, and if the different lengths impact on interpretation of data. Please explain the term “normalized cilia length” in Fig. 2E.

Previous studies have confirmed that 5HT₆ expression increases cilia length without noticeably affecting either ciliation efficiency or cilia morphology (Lin *et al.*, Nat Chem Biol, 2013; Su *et al.*, Nat Method, 2013; Phua *et al.*, Cell, 2017). Therefore, cilia length in cells expressing 5HT₆ can reach $\sim 10 \mu\text{m}$ (Figs. 1C, 2D). The length of cilia labeled by glutamylated tubulin is $\sim 2 \mu\text{m}$ (Fig. 2e, f, Supplementary Figs. 1,3,4). The length of Arl13B-labeled cilia is $\sim 2.5 \mu\text{m}$ (Fig. 2b, c, Supplementary Figs. 1,3,4). For the long cilia in 8XGBS-GFP cells, the cells were starved for 24 h for ciliogenesis and then incubated with rapamycin for 30 min and SAG in serum-free medium

for another 24 h. In our pilot experiment with 8XGBS-GFP cells, 2 days of serum starvation increased cilia length (see the figure below).

For the case of Figure 2e, to emphasize the change in each cilium at the same scale, the length of each cilium was normalized by dividing by the length value before rapamycin treatment. In the revised figure legend, we now describe how cilia length was normalized.

3. Velocities in Fig. 3 to be denoted as “ $\mu\text{m/s}$ ” not “ $\text{s}/\mu\text{m}$ ”

We have corrected it.

4. Writing needs to be improved throughout the text. Some examples include (a) rewriting parts of the abstract, (b) Page 11 (top line, add “alteration in” cilia length), Page 16 (change “The Lechtreck group once showed”), etc.

Following the reviewer’s suggestion, we had a professional proofreading company thoroughly re-edit the entire manuscript, along with rigorous clarification, and removal of overstated expressions.

5. Methods must include plasmids and/or if P2A-based co-expressing construct was used in all experiments. This is important for data interpretations, as equivalent amounts of FIB-FRB fusions are needed for the experiments.

We now describe the use of P2A-based constructs in the relevant figure legends.

6. Legends should include transfection constructs, cell numbers or confluence before starvation etc for the sake of reproducing the data by other labs.

We now describe the experimental protocols in the relevant figure legends.

7. Methods should mention % of transfected cells in 3T3 cells, and how transfected cells were selected for data collection (in many cases data collection being from <10 total cells from 3-4 experiments).

In most of our experiments, cells needed to be co-transfected with two or three constructs simultaneously. The co-transfection rate in 3T3 cells is 2~5%. Compared with the immunostaining assay, we usually obtained fewer samples in live-cell imaging experiments. The short interval of imaging with multiple channels and z-stacks allowed us to image only 2~3 fields

simultaneously. Under this condition, we usually obtained 8~10 cells from 3~4 independent experiments.

8. Probably, normalization of the fluorescent intensities of CCP5CD-FKBP signal with respect to the MAP-FRB signal is necessary to interpret the data in most figures.

We have now normalized the fluorescence intensities of YFP-FKBP, Neon-mGID1, and CCP5CD-Neon-FKBP by dividing the intensity values by that measured for the corresponding MAP4m (Fig. 1d and Supplementary Fig. 5). Because the value for the MAP4m signal remained essentially consistent in these experiments, the normalized kinetics of FKBP-tagged proteins going onto the axoneme is compared to the value measured by previous method.

Dear Editors and Reviewers,

We thank the editors and reviewers for the positive notes and constructive suggestions. We have made point-by-point responses to address reviewers' queries as below.

REVIEWERS' COMMENTS:

Reviewer #1 (Remarks to the Author):

In their manuscript, the authors develop a methodology to direct the deglutamylase CCP5 specifically into primary cilia. This system enables the alteration of the tubulin modification polyglutamylation specifically in, even after completion of ciliogenesis, and enables the understanding of the role of different PTMs in ciliary functions. Using this system, the authors show that deglutamylation of alters the intraflagellar transport, especially in the anterograde transport, and affects Hedgehog signaling.

Following the first round of review, the authors have addressed all the queries raised by the reviewer. They have made all suggested textual changes, and performed new experiments to confirm the data they had presented in the earlier version of the manuscript.

The novel version of the manuscript has thus strongly improved, and should be suitable for publication.

Thank you for the positive comments on our new version of the manuscript.

Reviewer #2 (Remarks to the Author):

The manuscript by Hong et al is greatly improved, particularly in the writing. Although the effects on cilia length, IFT and Hedgehog signaling are rather minor when the deglutamylase is recruited to the cilium, the method will be of interest to those in the field. I have a few suggestions for further improvements to the manuscript.

Thank you for positive comments on the utility of our work and offering constructive suggestions which help us improve our manuscript greatly.

1. Although the writing is more clear, it is a bit “robotic” at times. The following sentences are awkward and need to be rewritten:

-The first sentence of the Abstract. I don't understand what is meant by “The ciliary axoneme serves as the foundation for material transfer...”. The axoneme is the structural basis of the cilium. Yes, IFT transfers materials along the axoneme but the transition zone transfers material into cilia.

In order to shorten the abstract to fit the formatting requirements, we have removed the sentence “The ciliary axoneme serves as the ...”.

-The sentence on p. 4: “As a result, it has been shown that microtubules consisting of tubulins conjugated to glutamates to the C-terminal tail exhibit altered velocity and processivity of molecular motors

We have changed the original sentence to “As a result, it has been shown that chemical conjugation of glutamate side chains on purified microtubules increases the processivity and velocity of Kinesin-2 motors”.

-The sentence on p. 10: “When expressed in cells as a fusion protein of Neon-FKBP, both of which were localized to the cytosol (Supplementary Fig. 3b).“

We have changed the original sentence to “When expressed in cells as fusion proteins of Neon-FKBP, these two CCP5 truncations both localize in cytosol (Supplementary Fig. 3b)”.

-On p. 10: “To directly assess enzymatic activities of these CCP5 truncates“ the word truncates is unusual.

We have changed the word “truncates” to “truncations”.

-in the figure legends, it seems that IF should stand for immunofluorescence rather than immunostaining

We have changed “immunostaining” to “immunofluorescence assay” or “immunofluorescence staining”.

2. More experimental information is needed.

-What are the amino acids used in each of the CCP5 truncations and what mutations were introduced to generate the catalytically inactive CCP5CDDM construct?

We have added the details of CCP5 truncations in manuscript as below.

“Therefore, we assessed subcellular localization of truncation mutants of CCP5, namely N-terminus (CCP5N, residues 1-160) and catalytic domain (CCP5CD, residues 161-531). When expressed in cells as a fusion protein of Neon-FKBP, these two CCP5 truncations both localize in cytosol (Supplementary Fig. 3b). For further characterization of these truncation mutants, we introduced two point mutations (H252S and E255Q) to CCP5CD (CCP5CDDM) to impair the deglutamylation activity⁹.”

- what are the P2A-based constructs (p. 11)?

We have added the details of P2A in manuscript as below.

We inserted a viral P2A self-cleaving linker in between FKBP and FRB pieces, which encode relatively equal amounts of two proteins in subsequent experiment.

-which anti-glutamylated tubulin antibody was used in Figure 1?

The details of anti-glutamylated tubulin antibody were described in Methods.

Anti-glutamylated tubulin (1:500 dilution; AG-20B-0020, AdipoGen).

3. Two recent manuscripts from the Barr lab (Silva et al 2017, Curr Biol, O'Hagan et al 2017 Curr Biol) describe how changes in polyglutamylation impact axoneme structure and IFT in worms. A comparison to the work in these manuscripts is warranted for the discussion.

Thank you for the suggestions. We have discussed and cited these two important references in our manuscript.

4. The Discussion lacks an integration of the author's findings on Hedgehog signaling into the literature of the field.

We have added a paragraph to discuss our finding on Hedgehog signaling.

By verifying the effect of deglutamylation on the distribution of various Hh signaling

molecules as well as Gli transcriptional activities, we claimed that axonemal glutamylation positively regulates Hh signaling presumably through anterograde IFT-dependent mechanism (Fig. 7 and Supplementary Figs. 13 and 14). Our results showed that axonemal deglutamylation attenuates Hh signaling mainly by disturbing the ciliary entry of Smo and Gli proteins instead of the removal of their upstream negative regulators from cilia (Fig. 7 and Supplementary Figs. 13 and 14). Previous studies found that ciliary entry of Gli protein is driven by anterograde IFT, which offers a legitimate explanation on how deglutamylation-induced defects in anterograde IFT inhibit Hh signaling^{49,52}. However, several studies using pulse-chase assay and single molecule imaging have demonstrated that ciliary entry of Smo depends on lateral diffusion rather than anterograde IFT^{65,66}. Further work is required to decipher whether kinesin 2 assists Smo in crossing the diffusion barrier at cilia base^{48,67}.

Reviewer #3 (Remarks to the Author):

Thank you for your commentary on our work.

I am still not convinced with the quality of the IFT movies, and the minor increases in Gli1 levels in nucleus in control cells (Figure 7f). The authors have responded to most of my other comments satisfactorily.

** See Nature Research's author and referees' website at www.nature.com/authors for information about policies, services and author benefits

This email has been sent through the Springer Nature Tracking System NY-610A-NPG&MTS

Confidentiality Statement:

This e-mail is confidential and subject to copyright. Any unauthorised use or disclosure of its contents is prohibited. If you have received this email in error please notify our Manuscript Tracking System Helpdesk team at <http://platformsupport.nature.com>.

Details of the confidentiality and pre-publicity policy may be found here <http://www.nature.com/authors/policies/confidentiality.html>

Privacy Policy | Update Profile